# Learning to reconstruct signals from binary measurements alone

**Julián Tachella**                                              *julian.tachella@cnrs.fr*
*Physics Laboratory*
*CNRS & École Normale Supérieure de Lyon*

**Laurent Jacques**                                          *laurent.jacques@uclouvain.be*
*ICTEAM*
*UCLouvain*

**Reviewed on OpenReview:** *https: // openreview. net/ forum? id= ioFIAQOBOS*

## Abstract

Recent advances in unsupervised learning have highlighted the possibility of learning to reconstruct signals from noisy and incomplete linear measurements alone. These methods play a key role in medical and scientific imaging and sensing, where ground truth data is often scarce or difficult to obtain. However, in practice measurements are not only noisy and incomplete but also quantized. Here we explore the extreme case of learning from binary observations and provide necessary and sufficient conditions on the number of measurements required for identifying a set of signals from incomplete binary data. Our results are complementary to existing bounds on signal recovery from binary measurements. Furthermore, we introduce a novel self-supervised learning approach, which we name SSBM, that only requires binary data for training. We demonstrate in a series of experiments with real datasets that SSBM performs on par with supervised learning and outperforms sparse reconstruction methods with a fixed wavelet basis by a large margin.

## 1 Introduction

Continuous signals have to be quantized in order to be represented digitally with a limited number of bits in a computer. In many real-world applications, such as radar (Alberti et al., 1991), wireless sensor networks (Chen & Wu, 2015), and recommender systems (Davenport et al., 2014), the measured data is quantized with just a few bits per observation. The extreme case of quantization corresponds to observing a single bit per measurement. For example, single-photon detectors record the presence or absence of photons at each measurement cycle (Kirmani et al., 2014), and recommendation systems often observe a binary measurement of users' preferences only (*e.g.*, via thumbs up or down).

The binary sensing problem is formalized as follows: we observe binary measurements $y \in \{-1, 1\}^m$ of a signal $x \in \mathcal{X} \subset \mathbb{S}^{n-1}$ with unit norm[1] via the following forward model

$$y = \text{sign}\,(Ax) \tag{1}$$

where $A \in \mathbb{R}^{m \times n}$ is a linear forward operator. Recovering the signal from the measurements is an ill-posed inverse problem since there are many signals $x \in \mathbb{S}^{n-1}$ that are consistent with a given measurement vector $y$. Moreover, often the measurement matrix is incomplete $m < n$, *e.g.*, as in one-bit compressed sensing (Jacques et al., 2013), which makes the signal recovery problem even more challenging.

It is possible to obtain a good estimation of $x$ despite the binary quantization, if the set of plausible signals $\mathcal{X}$ is low-dimensional (Bourrier et al., 2014), *i.e.*, if it occupies a small portion of the ambient space $\mathbb{S}^{n-1}$.

---

[1]Note that the sensing model in (1) provides no information about the norm of $x$, so it is commonly assumed that signals verify $\|x\| = 1$.

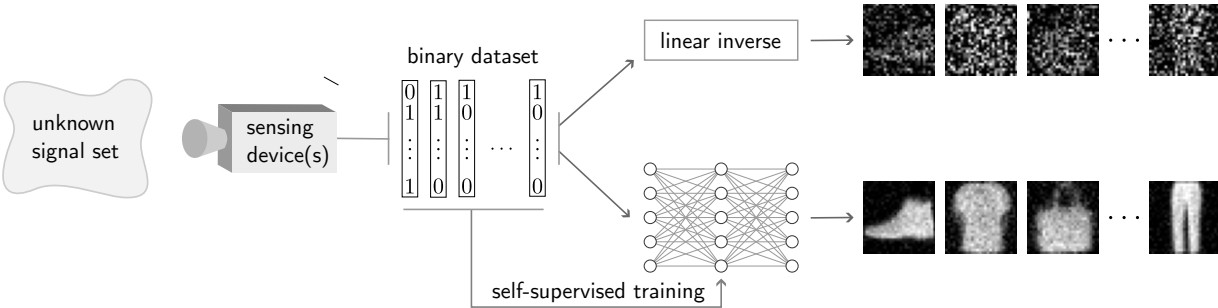

Figure 1: We propose a method for learning to reconstruct binary measurement observations, using only the binary observations themselves for training. The learned reconstruction function can discover unseen patterns in the data (in this case the clothes of fashionMNIST - see the experiments in Section 5), which cannot be recognized in the standard linear reconstructions (no learning). We also provide theoretical bounds that characterize how well we can expect to learn the set of signals from binary measurement data alone.

A popular approach is to assume that $\mathcal{X}$ is a single linear subspace or a union of subspaces (Jacques et al., 2013), imposing sparsity over a known dictionary. For example, the well-known total variation regularization assumes that the gradients of the signal are sparse (Rudin et al., 1992). However, in real-world settings, the set of signals $\mathcal{X}$ is generally unknown, and sparsity assumptions on an arbitrary dictionary yield a loose description of the true set $\mathcal{X}$, negatively impacting the quality of reconstructions obtained under this assumption. This limitation can be overcome by learning the reconstruction mapping $y \mapsto x$ (*e.g.*, with a deep neural network) directly from $N$ pairs of measurements and associated signals—*i.e.*, a supervised learning scenario with a labeled dataset $\{(y_i, x_i)\}_{i=1}^{N}$ with $N$ assumed sufficiently large. While this learning-based approach generally obtains state-of-the-art performance, it is often impractical since it can be very expensive or even impossible to obtain ground-truth signals $x_i$ for training. For example, recommender systems generally do not have access to high-resolution user ratings on all items for training.

In this paper, we investigate the problems of identifying the signal set and learning reconstruction mapping using a dataset of binary measurements only $\{y_i\}_{i=1}^{N}$. In this setting, if the measurement process is incomplete $m < n$, the matrix $A$ has a non-trivial nullspace and there is no information in the measurement data about the set of signals $\mathcal{X}$ in the nullspace (Chen et al., 2021). As a consequence, there is not enough information for learning the reconstruction function either. For example, the trivial pseudo-inverse reconstruction $f(y) = A^{\top}(AA^{\top})^{-1}y$ is perfectly consistent with the binary measurements, *i.e.*, $\mathrm{sign}\,(Af(y)) = y$, but is generally far from being optimal (Boufounos et al., 2015).

Here we show that it is still possible to (approximately) identify the signal set and learn to reconstruct the binary measurements, if the measurement operator varies across observations, *i.e.*,

$$y_i = \mathrm{sign}\,(A_{g_i} x_i) \tag{2}$$

where each signal $x_i$ is observed via one out of $G$ operators $g_i \in \{1, \ldots, G\}$, and $i = 1, \ldots, N$. This sensing assumption holds in various practical applications, where signals are observed through different operators (*e.g.*, recommendation systems access ratings about a different set of items for each user) or through an operator which changes through time (*e.g.*, a sensor that changes its calibration). Moreover, this assumption is also valid for the case where we obtain binary measurements via a single operator $A$, but the set $\mathcal{X}$ is known to be invariant to a group of invertible transformations $\{T_g\}_{g=1}^{G}$, such as translations or rotations. The invariance of $\mathcal{X}$ provides access to measurements associated with a set of (implicit) operators $\{A_g = AT_g\}_{g=1}^{G}$, as we have that

$$y = \mathrm{sign}\,\left(AT_g T_g^{-1} x\right) = \mathrm{sign}\,(AT_g x') \tag{3}$$

with $x' = T_g^{-1}x \in \mathcal{X}$ for all $g = 1, \ldots, G$. This observation has been exploited to perform fully unsupervised learning on various linear inverse problems, such as magnetic resonance imaging and computed tomography (Chen et al., 2021; 2022; Tachella et al., 2023).

| Assumption on $\mathcal{X} \subseteq \mathbb{S}^{n-1}$ | None | None | boxdim $< k$ |
|---|---|---|---|
| Assumption on $A_g \in \mathbb{R}^{m \times n}$, $g \in [G]$ | rank $[A_1^\top, \ldots, A_G^\top] < n$ | None | Gaussian |
| Identification error bounds | $\delta > 1$ | $\delta \gtrsim \frac{n}{mG}$ | $\delta \lesssim \frac{k+n/G}{m} \log \frac{nm}{k+n/G}$ |
| Section | Section 3.1 | Section 3.1 | Section 3.2 |

Table 1: Summary of the global model identification error $\delta$ bounds presented in this paper. The identification error $\delta$ corresponds to the maximal error of the optimal estimation of the signal set from binary measurement data alone (see Definition 3.1). The bounds depend on the size of the signals $n$, the number of binary measurement operators $G$ with $m$ measurements, and the dimension of the signal set $k$.

The problem of recovering a signal from binary measurements under the assumption of a known signal set has been extensively studied in the literature (Goyal et al., 1998; Jacques et al., 2013; Oymak & Recht, 2015). These works provide practical bounds that characterize the recovery error as a function of the number of measurements $m$ for different classes of signal sets. However, they assume that the signal set is known (or that there is enough ground-truth training data to approximate it), which is not often the case in real-world scenarios. Here we investigate the best approximation of the signal set that can be obtained from the binary observations. This approximation lets us understand how well we can learn the reconstruction function from binary data. To the best of our knowledge, the model identification problem has not been yet addressed, and we aim to provide the first answers to this problem here. The main contributions of this paper are:

- We show that for any $G$ sensing matrices $A_1, \ldots, A_G \in \mathbb{R}^{m \times n}$ and any dataset size $N$, there exists a signal set whose identification error (precisely defined in Section 3) from binary measurements cannot decay faster than $\mathcal{O}(\frac{n}{mG})$ when $m$ increases.

- We prove that, if each operator $A_g$, $g \in \{1, \ldots, G\}$, has iid Gaussian entries (a standard construction in one-bit compressed sensing), it is possible to estimate a $k$-dimensional[2] signal set up to a global error of $\mathcal{O}(\frac{k+n/G}{m} \log \frac{nm}{k+n/G})$ with high probability.

- We determine the *sample complexity* of the related unsupervised learning problem, *i.e.*, we find that, for $G$ operators with Gaussian entries, the number of distinct binary observations for obtaining the best possible approximation of a $k$-dimensional signal set $\mathcal{X}$ is $N = \mathcal{O}\big(G(\frac{m\sqrt{n}}{k})^{5k}\big)$ with controlled probability, which reduces to $N = \mathcal{O}\big(G(\frac{m}{k})^k\big)$ if $\mathcal{X}$ is a union of $k$-dimensional subspaces.

- We introduce a Self-Supervised learning loss for training reconstruction networks from Binary Measurement data alone (SSBM), and show experimentally that the learned reconstruction function outperforms classical binary iterative hard thresholding (Jacques et al., 2013) and performs on par with fully supervised learning on various real datasets.

A summary of the model identification bounds presented in this paper is shown in Table 1.

**Related Work**

**Unsupervised learning in inverse problems.** Despite providing very competitive results, most deep learning-based solvers require a supervised learning scenario, *i.e.*, they need measurements and signal pairs $\{(y_i, x_i)\}$, a labeled dataset, in order to learn the reconstruction function $y \mapsto x$. A first step to overcome this limitation is due to Noise2Noise (Lehtinen et al., 2018), where the authors show that it is possible to learn from only noisy data if two noisy realizations of the same signal $\{(x_i + n_i, x_i + n_i')\}$ are available for training. This approach has been extended to linear inverse problems with pairs of measurements $\{(A_{g_i} x_i + n_i, A_{g_i'} x_i + n_i')\}$ (Yaman et al., 2020; Liu et al., 2020). The equivariant imaging framework (Chen et al., 2021; 2022) shows that learning the reconstruction function from unpaired measurement data $\{Ax_i + n_i\}$ of a single incomplete linear operator $A$ is possible if the signal model is invariant to a group of transformations. This approach can also be adapted to the case where the signal model is not invariant, but measurements are

---
[2]The definition of dimension used in this paper is the upper box-counting dimension defined in Section 2.

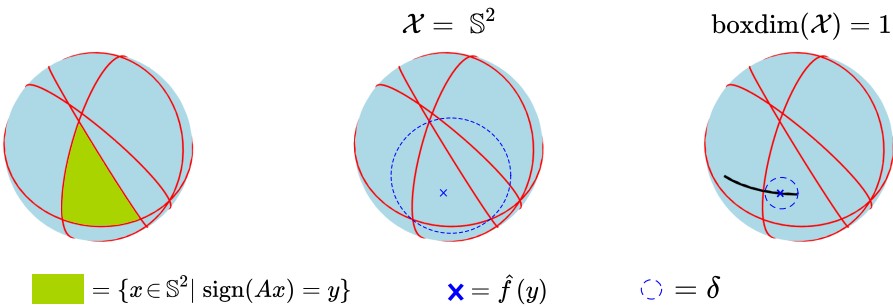

Figure 2: Geometry of the 1-bit signal recovery problem with $m = 5$ and $n = 3$. **Left:** The binary sensing operator $\text{sign}(A\cdot)$ defines a tessellation of the sphere into multiple *consistency cells*, which are defined as all vectors $x \in \mathbb{S}^2$ associated with the same binary code. The consistency cell associated with a given measurement $y$ is shown in green. Each red line is a great circle defined by all points of $\mathbb{S}^2$ perpendicular to one row of $A$. **Middle:** If the signal set consists of all vectors in the sphere, *i.e.*, $\mathcal{X} = \mathbb{S}^2$, the center of the cell is the optimal reconstruction $\hat{f}(y)$ (depicted with a blue cross) and the recovery error (denoted by $\delta$) is given by the radius of the cell. **Right:** If the signal set (depicted in black) occupies only a small subset of $\mathbb{S}^2$, *i.e.*, it has a small box-counting dimension, the optimal reconstruction corresponds to the center of the intersection between the signal set and the consistency cell, and the resulting signal recovery error is smaller.

obtained via many different operators $\{A_{g_i} x_i + n_i\}$ (Tachella et al., 2022). Necessary and sufficient conditions for learning in these settings are presented in Tachella et al. (2023), however under the assumption of linear observations (no quantization). Here we extend these results to the non-linear binary sensing problem with an unsupervised dataset with multiple operators $\{\text{sign}(A_{g_i} x_i)\}$ and $g_i \in \{1, \ldots, G\}$, or with a single operator and a group-invariant signal set $\{\text{sign}(A x_i)\}$.

**Quantized and one-bit sensing.** Reconstructing signals from one-bit compressive measurements is a well-studied problem (Goyal et al., 1998; Jacques et al., 2013; Oymak & Recht, 2015; Baraniuk et al., 2017), both in the (over)complete case $m \geq n$ (Goyal et al., 1998), and in the incomplete setting $m < n$, either under the assumption that the signals are sparse (Jacques et al., 2013), or more generally, that the signal set has small Gaussian width (Oymak & Recht, 2015). Some of these results are summarized in Section 2. The theoretical bounds presented in this paper complement those of signal recovery bounds from quantized data, as they characterize the fundamental limitations of model identification from binary measurement data.

**One-bit matrix completion and dictionary learning.** Matrix completion consists of inferring missing entries of a data matrix $Y = [y_1, \ldots, y_N]$, whose columns can be seen as partial observations of signals $x_i$, *i.e.*, $y_i = \text{sign}(A_{g_i} x_i)$ where the operators $A_{g_i}$ select a random subset of $m$ entries of the signal $x_i$. In order to recover the missing entries, it is generally assumed that the signals $x_i$ (the columns of $X = [x_1, \ldots, x_N]$) belong to a $k$-dimensional subspace with $k \ll n$. Davenport et al. (2014) solve this learning problem via convex programming and present theoretical bounds for the reconstruction error.

Zayyani et al. (2015) present an algorithm that learns a dictionary (*i.e.*, a union of $k$-dimensional subspaces) from binary data alone in the overcomplete regime $m > n$. Rencker et al. (2019) presents a similar dictionary learning algorithm with convergence guarantees. In this paper, we characterize the model identification error for the larger class of low-dimensional signal sets, which includes subspaces and the union of subspaces as special cases. Moreover, we propose a self-supervised method that learns the reconstruction mapping directly, avoiding an explicit definition (*e.g.*, a dictionary) of the signal set.

## 2 Signal Recovery Preliminaries

We begin with some basic definitions related to the one-bit sensing problem. The diameter of a set is defined as $\mathrm{diam}(S) = \sup_{u,v \in S} \|u - v\|$, and the radius is defined as half the diameter. Each row $a_i \in \mathbb{R}^n$ in the operator $A$ divides the unit sphere $\mathbb{S}^{n-1}$ into two hemispheres, *i.e.*, $\{x \in \mathbb{S}^{n-1} : a_i^\top x \geq 0\}$ and $\{x \in \mathbb{S}^{n-1} : a_i^\top x < 0\}$. Considering all rows, the operator $\mathrm{sign}\,(A \cdot)$ defines a *tesselation* of $\mathbb{S}^{n-1}$ into *consistency cells*, where each cell is composed of all the signals that are associated with a binary code $y$, *i.e.*, $\{x \in \mathbb{S}^{n-1} : \mathrm{sign}\,(Ax) = y\}$. The radius and number of consistency cells play an important role in the analysis of signal recovery and model identification. Figure 2 illustrates the geometry of the problem for $n = 3$ and $m = 5$.

The problem of recovering a signal from one-bit compressed measurements with a known signal set has been well studied (Goyal et al., 1998; Jacques et al., 2013; Oymak & Recht, 2015; Baraniuk et al., 2017). These works characterize the maximum estimation error across all signals obtained by an optimal reconstruction function $\hat{f}$, *i.e.*,

$$\delta = \max_{x \in \mathcal{X}} \|x - \hat{f}(\mathrm{sign}\,(Ax))\| \tag{4}$$

as a function of the number of measurements and complexity of the signal model. From a geometric viewpoint (see Figure 2), the optimal reconstruction function with respect to the norm $\|\cdot\|$ is given by the centroid (with respect to the same norm $\|\cdot\|$) of the intersection between the consistency cell associated with the measurement $y = y(x) = \mathrm{sign}\,(Ax)$, *i.e.*, $S_y := \{u \in \mathbb{S}^{n-1} : y = \mathrm{sign}\,(Au)\}$, and the signal set $\mathcal{X}$, *i.e.*,

$$\hat{f}(y) = \mathrm{centroid}(S_y \cap \mathcal{X}). \tag{5}$$

while the maximum reconstruction error is given by the intersection with maximal radius, that is

$$\delta = \max_{x \in \mathcal{X}} \mathrm{radius}(S_{y(x)} \cap \mathcal{X}). \tag{6}$$

In the overcomplete case $m > n$, assuming that all unit vectors are plausible signals, *i.e.*, $\mathcal{X} = \mathbb{S}^{n-1}$, the mean reconstruction error $\delta$ is given by the consistency cell with maximal radius, which scales as $\frac{n}{m}$ (see Proposition 5). The optimal rate is achieved by measurement consistent reconstruction functions, *i.e.*, those verifying $y = \mathrm{sign}\,(Af(y))$ (Goyal et al., 1998).

In the incomplete case $m < n$, non-trivial signal recovery is only possible if the set of signals occupies a low-dimensional subset of the unit sphere $\mathbb{S}^{n-1}$ (Oymak & Recht, 2015). For example, a common assumption is that $\mathcal{X}$ is the set of $k$-sparse vectors (Jacques et al., 2013). In this paper, we characterize the class of low-dimensional sets using a single intuitive descriptor, the box-counting dimension. The upper box-counting dimension (Falconer, 2004, Chapter 2) is defined for a compact subset $S \subset \mathbb{R}^n$ as

$$\mathrm{boxdim}\,(S) = \limsup_{\epsilon \to 0^+} \frac{\log \mathfrak{N}(S, \epsilon)}{\log 1/\epsilon} \tag{7}$$

where $\mathfrak{N}(S, \epsilon)$ is the minimum number of closed balls of radius $\epsilon$ with respect to the norm $\|\cdot\|$ that are required to cover $S$. This descriptor has been widely adopted in the inverse problems literature (Puy et al., 2017; Tachella et al., 2023), and it captures the complexity of various popular models, such as smooth manifolds (Baraniuk & Wakin, 2009) and union of subspaces (Blumensath & Davies, 2009; Baraniuk et al., 2017). For example, the set of $(k + 1)$-sparse vectors with unit norm has a box-counting dimension equal to $k$. The upper box-counting dimension is particularly useful to obtain an upper bound on the covering number of a set: if $\mathrm{boxdim}\,(\mathcal{X}) < k$, there exists a set-dependent constant $\epsilon_0 \in (0, \frac{1}{2})$ for which

$$\mathfrak{N}(\mathcal{X}, \epsilon) \leq \epsilon^{-k} \tag{8}$$

holds for all $\epsilon \leq \epsilon_0$ (Puy et al., 2017). The following theorem (proved in Appendix B) exploits this fact to provide a bound on the number of measurements needed for recovering a signal with an error smaller than $\delta$ from generic binary observations.



Figure 3: Illustration of the model identification problem from binary measurements with $n = 3$, $m = 4$, and $G = 3$. A signal set with box-counting dimension 1 is depicted in black. The red lines define the frontiers of the consistency cells associated with operators $A_1, \ldots, A_3$. **From left to right:** The signal set, the estimation of the signal set associated with $A_1, \ldots, A_3$ and the overall estimate $\hat{\mathcal{X}}$.

**Theorem 1.** *Let $A$ be a matrix with iid entries sampled from a standard Gaussian distribution and assume that* $\mathrm{boxdim}(\mathcal{X}) < k$, *such that* $\mathfrak{N}(\mathcal{X}, \epsilon) \leq \epsilon^{-k}$ *for all* $\epsilon < \epsilon_0$ *with* $\epsilon_0 \in (0, \frac{1}{2})$. *For* $\delta \leq \min\{30\sqrt{n}\epsilon_0, \frac{1}{2}\}$, *if the number of measurements verifies*

$$m \geq \tfrac{4}{\delta}\big(2k \log \tfrac{30\sqrt{n}}{\delta} + \log \tfrac{1}{\xi}\big) \tag{9}$$

*then for all $x, s \in \mathcal{X}$, we have that*

$$\mathrm{sign}\,(Ax) = \mathrm{sign}\,(As) \implies \|x - s\| < \delta \tag{10}$$

*with probability greater than $1 - \xi$.*

This result extends Theorem 2 in Jacques et al. (2013), which holds for $k$-sparse sets only, to general low-dimensional sets and is included in Appendix B. For example, if $\mathcal{X}$ is the intersection of $L$ $(s+1)$-dimensional subspaces with the unit sphere, Theorem 1 holds with constant $\epsilon_0 = (3^s L)^{-\frac{1}{k-s}}$ and $k > s$ (Vershynin, 2018, Chapter 4.2). This theorem tells us that we can recover sparse signals from binary measurements up to an error of

$$\mathcal{O}(\tfrac{k}{m} \log \tfrac{nm}{k})$$

which is sharp, up to the logarithmic factor (Jacques et al., 2013). Oymak and Recht (Oymak & Recht, 2015) present a similar result, stated in terms of the Gaussian width[3] of the signal set instead of the box-counting dimension.

## 3 Model Identification from Binary Observations

In this section, we study how well we can identify the signal set from binary measurement data associated with $G$ different measurement operators $A_1, \ldots, A_G \in \mathbb{R}^{m \times n}$. We focus on the problem of identifying the set $\mathcal{X}$ from the binary sets $\{\mathrm{sign}\,(A_g \mathcal{X})\}_{g=1}^G$. In practice, we observe a subset of each binary set $\mathrm{sign}\,(A_g \mathcal{X})$, however, in Section 3.4 we show that the number of elements in each of these sets is controlled by the box-counting dimension of $\mathcal{X}$, which is typically low in real-world settings (Hein & Audibert, 2005).

We start by analyzing how the different operators provide us with information about $\mathcal{X}$. Each forward operator $A_g$ constrains the signal space by the following set

$$\hat{\mathcal{X}}_g = \{v \in \mathbb{S}^{n-1} : \ \exists x_g \in \mathcal{X}, \ \mathrm{sign}\,(A_g v) = \mathrm{sign}\,(A_g x_g)\}. \tag{11}$$

Each set $\hat{\mathcal{X}}_g$ is thus composed of all unit vectors $v$ that are *consistent* with at least one point $x_g$ of $\mathcal{X}$ according to the binary mapping $\mathrm{sign}\,(A_g \cdot)$. We thus conclude that $\hat{\mathcal{X}}_g$ is essentially a *dilation* of $\mathcal{X}$—and we clearly have $\mathcal{X} \subset \hat{\mathcal{X}}_g$—whose extension is locally determined by specific cells of $\mathrm{sign}\,(A_g \cdot)$. A three-dimensional example with $m = 4$ measurements and $G = 3$ operators is presented in Figure 3. Note that, for a given binary mapping $\mathrm{sign}\,(A_g \cdot)$, each cell is characterized by one binary vector in the range of this mapping, so

---

[3]The Gaussian width of a set $S$ is defined as $\mathbb{E}_s\{\sup_{x \in S} x^\top s\}$ where $s$ is distributed as a standard Gaussian vector.

that, as shown in this figure, all cells provide a different tesselation of $\mathbb{S}^{n-1}$ whose size and dimension will play an important role in our analysis.

Since each $\hat{\mathcal{X}}_g$ is a dilation of $\mathcal{X}$, we can infer the signal set from the following intersection

$$\hat{\mathcal{X}} := \bigcap_{g=1}^{G} \hat{\mathcal{X}}_g, \tag{12}$$

which can be expressed concisely as

$$\hat{\mathcal{X}} = \left\{ v \in \mathbb{S}^{n-1} : \exists x_1, \ldots, x_G \in \mathcal{X}, \ \text{sign}\left(A_g v\right) = \text{sign}\left(A_g x_g\right), \ \forall g = 1, \ldots, G \right\}. \tag{13}$$

Due to the binary quantization, the inferred set will be larger than the true set, *i.e.*, $\mathcal{X} \subset \hat{\mathcal{X}}$. However, we will show that it is possible to learn a slightly *larger* signal set, defined in terms of a global identification error $\delta > 0$, *i.e.*, the open $\delta$-*tube*

$$\mathcal{X}_\delta = \{ v \in \mathbb{S}^{n-1} : \ \inf_{x \in \mathcal{X}} \|x - v\| < \delta \} \tag{14}$$

such that the inferred set is contained in it, *i.e.*, $\hat{\mathcal{X}} \subset \mathcal{X}_\delta$. We define the model identification error as the smallest $\delta$ such that $\hat{\mathcal{X}} \subset \mathcal{X}_\delta$ holds:

**Definition 3.1** (Model identification error)**.** The identification error of a signal set $\mathcal{X} \subset \mathbb{S}^{n-1}$ from binary sets $\{\text{sign}\left(A_g \mathcal{X}\right)\}_{g=1}^{G}$ is defined as $\min\{\delta \geq 0 : \hat{\mathcal{X}} \subseteq \mathcal{X}_\delta\}$.

For our developments to be valid, we will further assume that $\mathcal{X}$ is not too dense over $\mathbb{S}^{n-1}$ so that two tubes of $\mathcal{X}$ with two distinct radii are distinct.

**Assumption 1.** The set $\mathcal{X}$ is closed and there exists a maximal radius $0 < \delta_0 < 2$ for which $\mathcal{X}_\delta \subsetneq \mathcal{X}_{\delta_0}$ for any $0 < \delta < \delta_0$.

This assumption amounts to saying that there exists at least one open ball in $\mathbb{S}^{n-1}$ that does not belong to $\mathcal{X} \subset \mathbb{S}^{n-1}$. For instance, $\mathcal{X} = \mathbb{S}^{n-1}$ does not verify this assumption, and $\mathcal{X} = \mathbb{S}^{n-1} \cap \{x \in \mathbb{R}^n : x_1 \geq 0\}$ verifies it for $\delta_0 \leq \sqrt{2}$ since $\mathcal{X}_\delta = \mathbb{S}^{n-1}$ for any $\delta \geq \sqrt{2}$. The next subsections provide lower and upper bounds for $\delta$.

## 3.1 A Lower Bound on the Identification Error

We first aim to find a lower bound on the best $\delta$ achievable via the following oracle argument: if we had oracle access to $G$ measurements of each point $x$ in $\mathcal{X}$ through each of the $G$ different operators, we could stack them together to obtain a larger measurement operator, defined as

$$\begin{bmatrix} y_1 \\ \vdots \\ y_G \end{bmatrix} = \text{sign}\left(\bar{A} x\right) \ \text{with} \ \bar{A} = \begin{bmatrix} A_1 \\ \vdots \\ A_G \end{bmatrix} \in \mathbb{R}^{mG \times n}. \tag{15}$$

This oracle measurement operator provides a refined approximation of the signal set, specified as

$$\hat{\mathcal{X}}_{\text{oracle}} = \{ v \in \mathbb{S}^{n-1} : \ \exists x \in \mathcal{X}, \ \text{sign}\left(\bar{A} v\right) = \text{sign}\left(\bar{A} x\right) \}, \tag{16}$$

which is again a dilation of $\mathcal{X}$.

Figure 4 shows an example with the oracle set $\hat{\mathcal{X}}_{\text{oracle}}$, which provides a better (or equal) approximation of the signal set than (13), due to the fact that $\mathcal{X} \subset \hat{\mathcal{X}}_{\text{oracle}} \subseteq \hat{\mathcal{X}}$ by the construction of these sets. As the oracle estimate is composed of the cells associated with $\text{sign}\left(\bar{A}\cdot\right)$ which are intersected by the signal set, the oracle approximation error depends on the diameter of the intersected cells. Given a certain oracle tesselation of $\mathbb{S}^{n-1}$, the worst estimate of $\mathcal{X}$ is obtained when it intersects the largest cells in the tessellation. The following proposition formalizes the intuition that the maximum consistency cell diameter—*i.e.*, the greatest distance separating two binary consistent vectors of $\mathcal{X}$ according to $\bar{A}$—serves as a lower bound on the model identification error $\delta$.

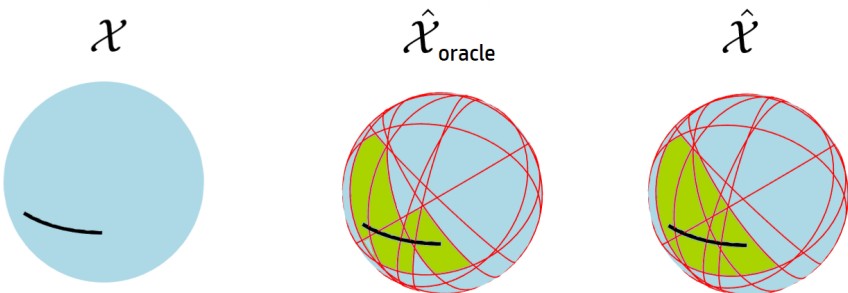

Figure 4: Illustration of the oracle argument in the example of Figure 3. **Left:** The signal set $\mathcal{X} \subset \mathbb{S}^2$ is depicted in black. **Middle:** Cells intersected by the oracle system are indicated in green. **Right:** The identified set $\hat{\mathcal{X}}$ is indicated in green, and *is larger* than the oracle counterpart.

**Proposition 2.** *Given $\bar{A} \in \mathbb{R}^{mG \times n}$, for any set $\mathcal{X} \subset \mathbb{S}^{n-1}$ respecting Assumption 1 with $0 < \delta_0 < 2$, there exists a rotation matrix $R \in SO(n)$ such that the rotated set*

$$\mathcal{X}' = \{v \in \mathbb{S}^{n-1} : v = Rx, \ x \in \mathcal{X}\} = R\mathcal{X} \tag{17}$$

*verifies $\hat{\mathcal{X}}'_{oracle} \not\subset \mathcal{X}'_\delta$ for any $\delta < \min\{d, \delta_0\}$ where $0 < d < 2$ is the largest cell diameter of the tesselation induced by $\mathrm{sign}\left(\bar{A}\cdot\right)$.*

*Proof.* Given $\delta < \delta_0$, the proof consists in choosing an appropriate rotation matrix, such that we can find a point $v$ which belongs to the oracle estimate $\hat{\mathcal{X}}'_{oracle}$ of the rotated set $\mathcal{X}'$, but doesn't belong to the $\delta$-tube $\mathcal{X}'_\delta$ of this set. From Assumption 1 and since the $\delta$-tube $\mathcal{X}_\delta$ is open, there exists $x \in \mathcal{X}$ and $v \notin \mathcal{X}_\delta$ such that $\|x - v\| = \delta$ Let $S$ denote the largest cell in the tesselation of $\mathbb{S}^{n-1}$ induced by $\mathrm{sign}\left(\bar{A}\cdot\right)$, such that $d = \mathrm{diam}(S)$. If $\delta < d$, we can always pick a rotation $R \in SO(n)$ such that both $x' = Rx$ and $v' = Rv$ belong to $S$. As $x' \in S$, $\mathcal{X}'$ intersects $S$ and we have that $S \subseteq \hat{\mathcal{X}}'_{oracle}$, and thus that $v' \in \hat{\mathcal{X}}'_{oracle}$. $\qquad\square$

In words, Proposition 2 shows that we can rotate any signal set $\mathcal{X}$ such that it intersects the largest consistency cell in the tesselation, obtaining a model identification error that is proportional to the maximum cell diameter. The rotation is used to remove the best-case scenarios where the signal set only intersects consistency cells that are smaller than the largest one.

In the rest of this subsection, we focus on bounding the maximum cell diameter, as it is directly related to the model identification error through Proposition 2. We start with the following proposition which shows that, if the stacked matrix is rank-deficient, all cells have the maximum possible diameter.

**Proposition 3.** *Consider the tessellation defined by $\mathrm{sign}\left(\bar{A}\cdot\right)$ with $\bar{A} \in \mathbb{R}^{mG \times n}$. If*

$$rank(\bar{A}) < n \tag{18}$$

*all the cells in the tessellation have a diameter equal to 2.*

*Proof.* If $\bar{A}$ has a rank smaller than $n$, it has a non-trivial nullspace. Let $v \in \mathbb{S}^{n-1}$ be an element in the nullspace with unit norm. Consider a cell associated with the code $\mathrm{sign}\left(\bar{A}x\right)$ for some $x \in \mathbb{R}^n$ inside the complement of this nullspace (*i.e.*, in the range of $\bar{A}^\top$). The points $\frac{x+v}{\|x+v\|}, \frac{x-v}{\|x-v\|} \in \mathbb{S}^{n-1}$ belong to this cell since they share the same code. As $\|x \pm v\| = \sqrt{\|v\|^2 + \|x\|^2}$ due to orthogonality, the distance between these two points is

$$\frac{2\|v\|}{\sqrt{\|v\|^2 + \|x\|^2}} = \frac{2}{\sqrt{1 + \|x\|^2}} \tag{19}$$

which tends to 2 as $\|x\|$ goes to zero, without modifying the cell code $\mathrm{sign}\left(\bar{A}x\right)$. $\qquad\square$

This result provides a practical necessary condition for model identification, which is summarized in the following corollary:

**Corollary 4.** *A necessary condition for the tesselation defined by* $\operatorname{sign}\left(\bar{A}\cdot\right)$ *to have consistency cells with a diameter smaller than 2 is that there are at least*

$$m \geq n/G$$

*measurements per operator.*

This proposition tells us that $n/G$ measurements are necessary in order to obtain non-trivial cell diameters, and thus to obtain a non-trivial estimation of $\mathcal{X}$.

Moreover, in practice, it is possible to compute the rank of the stacked matrix $\bar{A}$ via numerical approximations. The following theorem provides a more refined characterization of the oracle error for $m \geq n/G$:

**Proposition 5.** *Consider the tessellation defined by* $\operatorname{sign}\left(\bar{A}\cdot\right)$ *with* $\bar{A} \in \mathbb{R}^{mG \times n}$. *The largest cell in the tessellation has a diameter of size at least* $\frac{2}{3}\frac{n}{mG}$.

*Proof.* According to Thao & Vetterli (1996, Theorem A.7), the maximum number of cells $C_{\bar{A}}$ induced by a tessellation defined by $\operatorname{sign}\left(\bar{A}\cdot\right)$ with $\bar{A} \in \mathbb{R}^{mG \times n}$ can be upper bounded as

$$C_{\bar{A}} \leq \binom{mG}{n} 2^n.$$

As $\binom{mG}{n} \leq (\frac{emG}{n})^n$, we have that $C_{\bar{A}} \leq (\frac{2emG}{n})^n$. We can inscribe all cells into spherical caps $S_i$[4] of radius $\delta/2$, where $\delta$ is the maximum cell diameter. As shown in (Ball et al., 1997, Lemma 2.3), a spherical cap of radius $\delta/2$ has measure bounded by $\sigma(S_i) \leq (\frac{\delta}{4})^{n-1}\sigma_{n-1}$ where $\sigma_{n-1}$ is the measure of $\mathbb{S}^{n-1}$. Since the tessellation covers the unit sphere $\mathbb{S}^{n-1}$, we have that $\mathbb{S}^{n-1} \subseteq \cup_{i=1}^{C_{\bar{A}}} S_i$ and thus

$$\sum_{i=1}^{C_{\bar{A}}} \sigma(S_i) \geq \sigma_{n-1} \;\Rightarrow\; (\tfrac{2emG}{n})^n(\tfrac{\delta}{4})^n\sigma_{n-1} \geq \sigma_{n-1} \;\Rightarrow\; \delta \geq \tfrac{2}{3}\tfrac{n}{mG}.$$

$\square$

**Remark** The upper bound in this proposition is tight in the sense that there exist matrices that attain this bound. As a special case of Theorem 1 with $mG > n$ measurements and the box-counting dimension of $\mathcal{X}$ set to $n$, for an $mG \times n$ Gaussian random matrix $\bar{A}$, and by choosing the minimal number of required measurements in the condition of this theorem, we have with high probability that the largest cell of $\operatorname{sign}\left(\bar{A}\cdot\right)$ has a diameter that decays like $\mathcal{O}(\frac{n}{mG})$ up to log factors. By a standard boosting argument[5], it thus means that there exists a $mG \times n$ matrix $\bar{A}$ with the same consistency cell diameter decay.

As stated in the following corollary, Proposition 5 shows that the model identification error cannot decrease faster with the number of measurements and operators than $\mathcal{O}(\frac{n}{mG})$, since the largest cell in any oracle tesselation has a diameter of at least $\frac{2}{3}\frac{n}{mG}$.

**Corollary 6.** *Given $G$ operators $A_1, \ldots, A_G \in \mathbb{R}^{m \times n}$, and any set $\mathcal{X} \subset \mathbb{S}^{n-1}$ verifying Assumption 1 with $0 < \delta_0 < 2$, for any $0 < \delta < \min(\delta_0, \frac{2}{3}\frac{n}{mG})$, there exists a rotation $R$ such that the inferred signal set $\hat{\mathcal{X}}'$ of $\mathcal{X}' = R\mathcal{X}$ is not included in $\hat{\mathcal{X}}'_\delta$, i.e., $\hat{\mathcal{X}}' \not\subset \mathcal{X}'_\delta$.*

*Proof.* If $\mathcal{X} \subset \mathbb{S}^{n-1}$ respects Assumption 1 with $0 < \delta_0 < 2$, and $\hat{\mathcal{X}}_{\text{oracle}}$ and $\hat{\mathcal{X}}$ are the oracle set associated with $\bar{A}$ and the inferred set of $\mathcal{X}'$, respectively, then, as derived previously, we know that $\mathcal{X} \subset \hat{\mathcal{X}}_{\text{oracle}} \subset \hat{\mathcal{X}}$. According to Proposition 2 and Proposition 5, there exists a rotation $R$ such that $\hat{\mathcal{X}}'_{\text{oracle}} = R\hat{\mathcal{X}}_{\text{oracle}} \not\subset \mathcal{X}'_\delta = R\mathcal{X}_\delta$ for $0 < \delta < \min(\delta_0, \frac{2}{3}\frac{n}{mG})$. Therefore, from the inclusion above, we thus see that there exists $\hat{\mathcal{X}}' = R\hat{\mathcal{X}} \not\subset \mathcal{X}'_\delta$. $\square$

---

[4] A spherical cap of radius $r$ around a point $v \in \mathbb{S}^{n-1}$ is defined as $\{x \in \mathbb{S}^{n-1} : \|x - v\| < r\}$.

[5] Assuming a Gaussian matrix will fail to have the desired cell size with probability at most $\xi < 1$, the probability that 1 out of $r$ independent Gaussian matrices will fail to have this property is at least $1 - \xi^r$, which can be made arbitrarily high by increasing $r$.

At this stage, it is natural to ask if the condition in Corollary 4 is sufficient to upper bound the model identification error $\delta$. The answer is negative since, for certain families of operators, the maximum consistency cell associated with $\text{sign}\,(\bar{A}\cdot)$ does not decrease with the number of measurements $m$ or operators $G$, as illustrated by the following example inspired by a case considered in (Plan & Vershynin, 2013, Sec. 1.1).

**Example 3.1.** Consider the operators with binary[6] entries $A_1, \ldots, A_G \in \{-1, 1\}^{m \times n}$. Let $x_\lambda = e_1 + \lambda e_2 \in \mathbb{R}^n$ where $e_i \in \mathbb{S}^{n-1}$ is the $i$th canonical vector and $\lambda$ is a scalar. Due to the quantization of the operators, we have that $\text{sign}\,(\bar{A}x_\lambda) = \text{sign}\,(\bar{A}x_{\lambda'})$ for any value of $\lambda, \lambda' \in (-1, 1)$. Thus, there exists a cell in the oracle tesselation associated with $\text{sign}\,(\bar{A}\cdot)$ which contains the set of points $\{\frac{x_\lambda}{\|x_\lambda\|}\}_{\lambda \in (-1,1)}$ and thus has a diameter equal to $\sqrt{2}$, independently of the values of $m$ and $G$.

In the next subsection, we obtain an upper bound on the model identification error which overcomes this pathological example by sampling the operators from a continuous random distribution.

## 3.2 A Sufficient Condition for Model Identification

We now seek a sufficient condition on the number of measurements per operator that guarantees the identification of $\mathcal{X}$ up to a global error of $\delta$. As with the sufficient conditions ensuring signal recovery (see Section 2), we assume that $\mathcal{X}$ is low-dimensional to provide a bound that holds with high probability if the entries of the operators are sampled from a Gaussian distribution.

**Theorem 7.** *Given the operators $A_1, \ldots, A_G \in \mathbb{R}^{m \times n}$ with entries i.i.d. as a standard Gaussian distribution, a low-dimensional signal set $\mathcal{X}$, with $\text{boxdim}\,(\mathcal{X}) < k$, such that $\mathfrak{N}(\mathcal{X}, \epsilon) \leq \epsilon^{-k}$ for all $\epsilon < \epsilon_0$ with $\epsilon_0 \in (0, \frac{1}{2})$. For $0 < \delta \leq \min\{18\sqrt{n}\epsilon_0, 1\}$ and some failure probability $0 < \xi < 1$, if the number of measurements per operator verifies*

$$m \geq \tfrac{4}{\delta}\left[(k + \tfrac{n}{G})\log \tfrac{54\sqrt{n}}{\delta} + \tfrac{1}{G}\log \tfrac{1}{\xi}\right] \tag{20}$$

*then with probability at least $1 - \xi$, we have that $\hat{\mathcal{X}} \subseteq \mathcal{X}_\delta$.*

The proof is included in Appendix C. Theorem 7 provides a bound on $\delta$, *i.e.*, how precisely we can characterise the signal set $\mathcal{X}$, which we can compare with the lower bound in Proposition 5. From (20) we have that (see Appendix C for a detailed derivation),

$$\delta = \mathcal{O}(\tfrac{k+n/G}{m}\log\tfrac{nm}{k+n/G}). \tag{21}$$

The bound in (21) is consistent with existing model identification bounds in the linear setting (Tachella et al., 2023), which require $m > k + n/G$ measurements per operator for uniquely identifying the signal set.

## 3.3 Learning to Reconstruct

The best reconstruction function $\hat{f}$ that can be learned from binary measurements alone can be defined as a function of the identified set $\hat{\mathcal{X}}$, as defined in (5):

$$\hat{f}(y) = \text{centroid}(S_y \cap \hat{\mathcal{X}}) \tag{22}$$

for a binary input $y$ with associated consistency cell $S_y = \{v \in \mathbb{S}^{n-1}: \text{sign}\,(Av) = y\}$. The reconstruction error of $\hat{f}$ is lower bounded by the radius of the set $S_y \cap \hat{\mathcal{X}}$. This error must be larger than the error of a reconstruction function that has full knowledge about the signal set $\mathcal{X}$. Intuitively, if we have a large model identification error, $\hat{\mathcal{X}}$ will be a bad approximation of $\mathcal{X}$ and thus $\hat{f}$ will obtain large reconstruction errors. The following proposition formalizes this intuition, showing that the reconstruction error of $\hat{f}$ is lower bounded by the model identification error.

**Proposition 8.** *Given $G$ operators $A_1, \ldots, A_G \in \mathbb{R}^{m \times n}$ and a set $\mathcal{X} \subset \mathbb{S}^{n-1}$ with model identification error equal to $\delta$, there exist points $x_g \in \mathcal{X}$ for $g = 1, \ldots, G$ such that the reconstruction error is*

$$\|\hat{f}\,(\text{sign}\,(A_g x_g)) - x_g\| \geq \delta/2,$$

---

[6]This example can be generalized to operators with entries belonging to a discrete set $Q$, and show that there exist cells with diameter equal to $\sqrt{\Delta}$ where $\Delta$ is the minimum distance between two elements in $Q$.

where $\hat{f}$ is the optimal reconstruction function that can be learned from the measurement data $\{\text{sign}(A_g\mathcal{X})\}_{g=1}^G$, as defined in (22).

*Proof.* Following Definition 3.1 of model identification error, there exists a point $\hat{x} \in \hat{\mathcal{X}}$ such that $\|x - \hat{x}\| \geq \delta$ for all $x \in \mathcal{X}$. According to the construction of the inferred set $\hat{\mathcal{X}}$ in (13), there exist some $x_1, \ldots, x_G \in \mathcal{X}$ such that $\text{sign}(A_g\hat{x}) = \text{sign}(A_g x_g)$ for all $g = 1, \ldots, G$. Therefore, for any $g \in \{1, \ldots, G\}$, the diameter of the set $S_{\text{sign}(A_g x_g)} \cap \hat{\mathcal{X}}$ is at least $\|x_g - \hat{x}\|$ since both $x_g$ and $\hat{x}$ belong to this set. As the optimal reconstruction function outputs the centroid of the set (as defined in (22)), the reconstruction error of the point $x_g$ is at least $\|x_g - \hat{x}\|/2 \geq \delta/2$. $\qquad\square$

Therefore, we can use the results on model identification developed in Section 3.1 to lower bound the reconstruction error for the case where the function is learned from measurement data only. In particular, combining this result with Corollary 6, we obtain that the (worst-case) reconstruction error should be larger than $\frac{1}{3}\frac{n}{mG}$. It is worth noting that this result also holds for the case where we have a single operator and group invariance, *i.e.*, when $A_g = AT_g$ for $g = 1, \ldots, G$.

An upper bound on the reconstruction error is harder to obtain. Unfortunately, Theorems 1 and 7 do not automatically translate into a bound on the optimal reconstruction error of the reconstruction function defined in (22). Theorem 7 implies that the optimal unsupervised reconstruction $\hat{f}(\text{sign}(A_g x))$ is at most $\mathcal{O}(\frac{k+n/G}{m}\log\frac{nm}{k+n/G})$ away from the signal set $\mathcal{X}$, but does not guarantee that it is close to $x$. Nonetheless, we conjecture that this rate holds with high probability if the operators follow a Gaussian distribution:

**Conjecture 9.** *Given binary measurements from the operators $A_1, \ldots, A_G \in \mathbb{R}^{m\times n}$ with entries i.i.d. from a standard Gaussian distribution, the optimal reconstruction function defined in (22) has a maximal reconstruction error that is upper bounded as $\mathcal{O}(\frac{k+n/G}{m}\log\frac{nm}{k+n/G})$ with high probability.*

Conjecture 9 hypothesizes that the optimal unsupervised reconstruction function should obtain a similar performance than the supervised one, *i.e.*, $\mathcal{O}(\frac{k}{m}\log\frac{nm}{k})$ shown in Theorem 1, if the number of operators is sufficiently large, *i.e.*, $G > n/k$. In the experiments in Section 5, we provide empirical evidence that supports this hypothesis.

### 3.4 Sample complexity

We end our theoretical analysis of the unsupervised learning problem by bounding its *sample complexity*, *i.e.*, we bound the number $N$ of *distinct* binary measurement vectors $\{y_i\}_{i=1}^N$ that must be acquired for obtaining the best approximation of the signal set $\mathcal{X}$ from binary data.

Since we observe binary vectors $y \in \{\pm1\}^m$, there is a limited number of different binary observations. We could naively expect to observe up to $2^m$ different vectors per measurement operator (*i.e.*, all possible binary codes with $m$ bits), requiring at most $N \leq G2^m$ samples to fully characterize the best approximation of the signal set $\hat{\mathcal{X}}$ defined in (13). Fortunately, as already exploited in the proof of Proposition 5, this upper bound can be significantly reduced if the signal set has a low box-counting dimension, as not all cells in the tessellation will be intersected by the signal set (see Figure 3). We can thus obtain a better upper bound by counting the number of intersected cells, denoted as $|\text{sign}(A\mathcal{X})|$.

If $\mathcal{X}$ is the intersection of a single $k$-dimensional subspace with the unit sphere, (Thao & Vetterli, 1996, Theorem A.7) tells us that, for any matrix $A \in \mathbb{R}^{m\times n}$ with $m \geq k$, there are $|\text{sign}(A\mathcal{X})| \leq 2^k\binom{m}{k}$ intersected cells. More generally, if $\mathcal{X}$ is a union of $L$ subspaces, we have $|\text{sign}(A\mathcal{X})| \leq L2^k\binom{m}{k}$. Thus, using the fact that $\binom{m}{k} \leq \left(\frac{3m}{k}\right)^k$, from the $G$ measurement operators, we can observe up to

$$N \leq GL(\tfrac{6m}{k})^k \tag{23}$$

different measurement vectors. However, this result only holds for a union of subspaces having each dimension $k$. The following theorem extends this result to more general low-dimensional sets with small upper box-counting dimension.

**Theorem 10.** *Let the entries of $A \in \mathbb{R}^{m \times n}$ be sampled from a standard Gaussian distribution, and let $\mathcal{X} \subseteq \mathbb{R}^n$ with* $\mathrm{boxdim}(\mathcal{X}) < k$. *If* $k/(m\sqrt{n}) < \min(\epsilon_0, 1/2)$, *then, in expectation, the cardinality of the measurement set is bounded as*

$$\mathbb{E}|\operatorname{sign}(A\mathcal{X})| \leq \left(\tfrac{em\sqrt{n}}{k}\right)^k. \tag{24}$$

*Moreover, given a failure probability $0 < \xi < 1$, if $2k/(m\sqrt{n}) \leq \min(\epsilon_0, 1/2)$, then, with probability $1 - \xi$, we have*

$$|\operatorname{sign}(A\mathcal{X})| \leq (\tfrac{1}{\xi})^4 \left(\tfrac{3m\sqrt{n}}{5k}\right)^{5k}. \tag{25}$$

The proof is included in Appendix D. This result depends on the square root of the ambient dimension $\sqrt{n}$ due to the application of Lemma 11 and can be suboptimal for some signal sets. For example, the bound in (23) avoids this dependency for the case where $\mathcal{X}$ is a union of subspaces.

In the setting where we observe measurements through $G$ independent forward operators, we sum the number of intersected cells for each operator, so that with probability exceeding $1 - G\xi$ for $0 < \xi < 1$ (by a union bound), the number of different binary measurement vectors is then bounded by

$$N = \mathcal{O}\Big(G\left(\tfrac{m\sqrt{n}}{k}\right)^{5k}\Big)$$

with a hidden multiplicative constant depending on $\xi$. Similarly to (23), this bound scales exponentially only in the model dimension $k$ but not in the number of measurements $m$ or operators $G$. In the setting of a single operator and a $k$-dimensional invariant signal set, we have the upper bound $N = \mathcal{O}\Big(\left(\tfrac{m\sqrt{n}}{k}\right)^{5k}\Big)$.

## 4 Learning Algorithms

In this section, we present a novel algorithm for learning the reconstruction function $f : (y, A) \mapsto x$ from $N$ binary measurement vectors $\{(y_i, A_{g_i})\}_{i=1}^N$, which is motivated by the analysis in Section 3. We parameterize the reconstruction function using a deep neural network with parameters $\theta \in \mathbb{R}^p$. The learned function can take into account the knowledge about the forward operator by simply applying a linear inverse at the first layer, *i.e.*, $f_\theta(y, A) = \tilde{f}_\theta(A^\top y)$, or using more complex unrolled optimization architectures (Monga et al., 2021).

In the case where we observe measurements associated with $G$ different forward operators, we propose the SSBM loss

$$\underset{\theta \in \mathbb{R}^p}{\arg\min} \sum_{i=1}^N \Big[ \mathcal{L}_{\mathrm{MC}}(y_i, A_{g_i}\hat{x}_{\theta,i}) + \alpha \sum_{s \neq g_i} \|\hat{x}_{\theta,i} - f_\theta(\operatorname{sign}(A_s\hat{x}_{\theta,i}), A_s)\|_2^2 \Big], \tag{26}$$

where $\hat{x}_{\theta,i} = f_\theta(y_i, A_{g_i})$, the cost $\mathcal{L}_{\mathrm{MC}}(y_i, A_{g_i}\hat{x}_{\theta,i}) \geq 0$ enforces *measurement consistency* (MC), *i.e.*, require that $y_i = \operatorname{sign}(A_{g_i}\hat{x}_{\theta,i})$, and $\alpha \in \mathbb{R}_+$ is a hyperparameter controlling the trade-off between the two terms involved. In the setting where we have a single operator and the set $\mathcal{X}$ is invariant to a group of transformations $\{T_g\}_{g=1}^G$ such as rotations or translations, we aim to learn a reconstruction function $f_\theta : y \mapsto x$ (we remove the dependence of $f_\theta$ on $A$ to simplify the notation) via the following self-supervised loss:

$$\underset{\theta \in \mathbb{R}^p}{\arg\min} \sum_{i=1}^N \Big[ \mathcal{L}_{\mathrm{MC}}(y_i, A\hat{x}_{\theta,i}) + \alpha \sum_{g=1}^G \|T_g\hat{x}_{\theta,i} - f_\theta(\operatorname{sign}(AT_g\hat{x}_{\theta,i}))\|_2^2 \Big], \tag{27}$$

where $\hat{x}_{\theta,i} = f_\theta(y_i)$ and $\alpha \in \mathbb{R}_+$. In practice, we minimize (26) by mini-batching approaches (*e.g.*, stochastic gradient descent) by using sampling one out of the $G$ operators at random per batch. In both cases, we choose the measurement consistency term to be the logistic loss, *i.e.*,

$$\mathcal{L}_{\mathrm{MC}}(y, \hat{y}) = \log(1 + \exp(-y \circ \hat{y})) \tag{28}$$

which enforces sign-consistent predictions which are far from zero, as the logistic function tends asymptotically towards zero as $|\hat{y}| \to \infty$. An empirical analysis in Section 5 shows that the logistic loss obtains the best performance across various popular consistency losses.

**Analysis of the proposed loss**  We focus on the multi-operator loss in (26), although a similar analysis also holds for the equivariant setting. The first term of the loss enforces measurement consistency, *i.e.*, requires $y_i = \text{sign}\left(A_{g_i} f_\theta(y_i, A_{g_i})\right)$ for every $y_i$ in the dataset. However, in the incomplete setting $m < n$, the simple pseudo-inverse solution

$$f(y, A_g) = A_g^\dagger y \tag{29}$$

with $A_g^\dagger = A_g^\top (A_g A_g^\top)^{-1}$, is measurement consistent for any number of operators $G$ and training data $N$. Therefore, the first loss does not prevent learning a function $f_\theta(y, A_g)$ which acts independently for each operator (as if there were $G$ independent learning problems). The second loss *bootstraps* the current estimates $\hat{x}_{i,\theta} = f_\theta(y_i, A_{g_i})$ as new ground truth references, mimicking the supervised loss

$$\sum_{i=1}^{N} \sum_{s=1}^{G} \|\hat{x}_{i,\theta} - f_\theta(\text{sign}\left(A_s \hat{x}_{i,\theta}\right), A_s)\|^2, \tag{30}$$

in order to enforce consistency across operators. Importantly, this additional loss avoids the trivial pseudo-inverse solution in (29), as

$$A_g^\dagger y - A_s^\dagger \text{sign}\left(A_s A_g^\dagger y\right) \neq 0 \tag{31}$$

for $g \neq s$ if the nullspaces of $A_g$ and $A_s$ are different, *e.g.*, if the necessary condition in Corollary 4 is verified.

**Model identification perspective**  The learning algorithm constructs a discrete approximation of the signal set using the reconstructed dataset, *i.e.*, $\cup_{g=1}^{G} \tilde{\mathcal{X}}_g$ where $\tilde{\mathcal{X}}_g = f_\theta(\mathcal{Y}_g, A_g)$ for $g = 1, \ldots, G$ and $\mathcal{Y}_g$ is the subset of measurement vectors associated with the $g$th operator. From a model identification perspective, the measurement consistency loss ensures that $\mathcal{Y}_g = A_g \tilde{\mathcal{X}}_g$ for all $g = 1, \ldots, G$. The second loss ensures consistency across all operators, *i.e.*, $\tilde{\mathcal{X}}_g = f_\theta(A_s \tilde{\mathcal{X}}_g, A_s)$ for all pairs $s, g \in \{1, \ldots, G\}$, acting as a proxy for $\tilde{\mathcal{X}}_g = \tilde{\mathcal{X}}_s$.

## 5 Experiments

For all experiments, we use measurement operators with entries sampled from a standard Gaussian distribution and evaluate the performance of the algorithms using by computing the average peak-to-signal ratio (PSNR) on a test set with $N'$ ground-truth signals, that is:

$$\frac{1}{N'} \sum_{i=1}^{N'} \text{PSNR}\left(x_i', f_\theta(\text{sign}\left(A_{g_i} x_i'\right), A_{g_i})\right), \tag{32}$$

where the PSNR is computed after normalizing the reconstructed image such that it has the same norm as the reference image, *i.e.*,

$$\text{PSNR}(x, \hat{x}) = -20 \log \|x - \hat{x}\frac{\|x\|}{\|\hat{x}\|}\|. \tag{33}$$

We choose $f_\theta(y, A) = \tilde{f}_\theta(A^\top y)$ where $\tilde{f}_\theta$ is the U-Net network used in (Chen et al., 2021) with weights $\theta$, and train for 400 epochs with the Adam optimizer with learning rate $10^{-4}$ and standard hyperparameters $\beta_1 = 0.9$ and $\beta_2 = 0.99$.

### 5.1 MNIST experiments

We evaluate the theoretical bounds using the MNIST dataset, which consists of greyscale images with $n = 784$ pixels and whose box-counting dimension is approximately $k \approx 12$ (Hein & Audibert, 2005). We use $6 \times 10^4$ images for training and $10^3$ for testing.

**Multiple operators setting.**  We start by comparing the logistic consistency loss in (28) with the following alternatives:

- Standard $\ell_p$-loss, $\mathcal{L}_{\text{MC}}(y, \hat{y}) = \|y - \hat{y}\|_p^p$. As this loss is zero only if $\hat{y} = y$, it promotes sign consistency, $\text{sign}(\hat{y}) = y$ and unit outputs $|\hat{y}| = 1$.

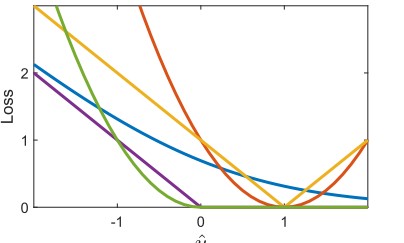 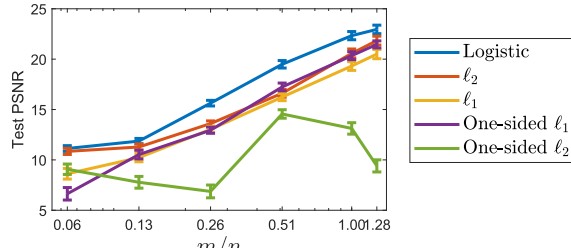

Figure 5: Evaluated training losses for enforcing sign measurement consistency $\operatorname{sign}(A\hat{x}) = y$ of reconstructions $f_\theta(y) = \hat{x}$. **Left:** The loss functions are shown for the case $y = 1$. **Right:** Average test PSNR of different measurement consistency losses on the MNIST dataset with $G = 10$ operators.

- One-sided $\ell_p$-loss, $\mathcal{L}_{\mathrm{MC}}(y, \hat{y}) = \|\max(-y \circ \hat{y}, 0)\|_p^p$ where $\circ$ denotes element-wise multiplication and the max operation is performed element-wise. This loss is zero as long as $\operatorname{sign}(\hat{y}) = y$ regardless of the value of $|\hat{y}|$.

For all losses except the logistic loss, setting the trade-off parameter $\alpha = 1$ obtained best results. For the logistic loss, we performed a sweep over different values of $\alpha$ and $m$, finding that the optimal choice of $\alpha$ decreases with $m$ (see Appendix E for more details). Thus, we set $\alpha = 0.1$ for $m < n$ and $\alpha = 0.06$ for $m \geq n$. Figure 5 shows the different losses and the test performance for different values of measurements using $G = 10$ operators. The logistic loss obtains the best performance across all sampling regimes, whereas the one-sided $\ell_2$ loss obtains the worst results.

Secondly, we compare the logistic loss with the following learning schemes:

- Linear inverse (no learning), defined as $\hat{x}_i = A_{g_i}^\top y_i$. This reconstruction can fail to be measurement consistent (Goyal et al., 1998).

- Standard supervised learning loss, defined as $\sum_{i=1}^{N} \|x_i - f_\theta(y_i, A_{g_i})\|^2$. We also evaluate this loss together with the cross-operator consistency term in (26) which we denote as supervised+.

- Measurement consistency loss, defined as $\sum_{i=1}^{N} \mathcal{L}_{\mathrm{MC}}(y_i, A_{g_i} f_\theta(y_i, A_{g_i}))$ using the logistic loss.

- The binary iterative hard-thresholding (BIHT) reconstruction algorithm (Jacques et al., 2013) with a Daubechies4 orthonormal wavelet basis. The step size and sparsity level of the algorithm were chosen via grid search. It is worth noting that the best-performing sparsity level increases as the number of measurements $m$ is increased.

- Proposed SSBM loss in (26) using the logistic loss for measurement consistency.

Test PSNR values obtained for the case of $G = 10$ operators are shown in the left subfigure of Figure 6, where the PSNR in dB is plotted against $m/n$ in log-scale representation. The measurement consistency approach obtains performance similar to simply applying a linear inverse for the incomplete $m/n < 1$ setting, whereas it obtains a significant improvement over the linear inverse in the overcomplete case $m/n \geq 1$. This gap can be attributed to the lack of measurement consistency of the linear reconstruction algorithm (Goyal et al., 1998). The proposed loss obtains a performance that is several dBs above the linear inverse and BIHT for all sampling regimes. BIHT relies on the wavelet sparsity prior, which does not capture well enough the MNIST digits. SSBM performs similarly to supervised learning as the sampling ratio tends to 1, and perhaps surprisingly, it obtains slightly better performance than supervised learning for $m/n = 1.28$. However, adding the cross-operator consistency loss to the supervised method (*i.e.*, the method supervised+ in Section 5.1) performs better for all sampling regimes than SSBM.

The right plot in Figure 6 compares the performance of the SSBM with the bounds in Proposition 8 and Conjecture 9. These bounds behave almost linearly in this log-log plot of both the error—through the PSNR—and

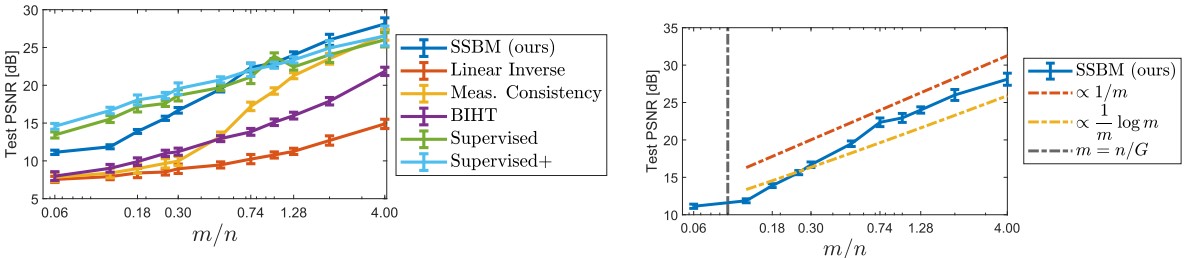

Figure 6: **Left:** Average test PSNR of different supervised and unsupervised algorithms on the MNIST dataset with $G = 10$ operators. **Right:** The performance of the SSBM method follows closely the bounds in Conjecture 9.

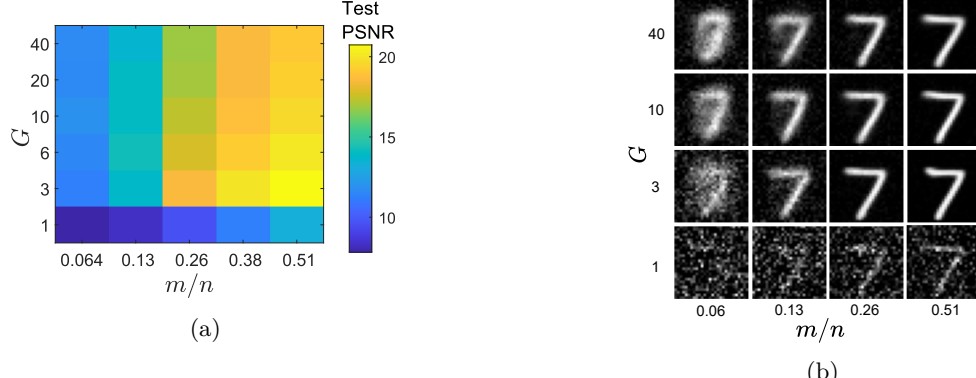

Figure 7: (**a**) Average test PSNR and (**b**) reconstructed test images of the proposed unsupervised method for different numbers of operators $G$ and measurements $m$.

the log-scale representation of $m/n$. We thus observe a good agreement between the predictions in Conjecture 9 and the performance in practice.

Figure 7 shows the average test PSNR and reconstructed images obtained by the proposed self-supervised method for different values of $G$ and $m$. The method fails to obtain good reconstructions when $G = 1$, as the necessary condition in Corollary 4 is not fulfilled.

**Noisy measurements** In many sensing applications, the sensor data are subject to noise. In the setting of binary measurements, noise affects the measurements by flipping the sign, as the observations can only be $-1$ or $+1$. We evaluate the proposed algorithm with $m = 274$ measurements and $G = 10$ operators and different noise levels, according to the model

$$y_i = \text{sign}\left(A_{g_i,i}x_i + \epsilon_i\right) \tag{34}$$

where $\epsilon_i \sim \mathcal{N}(0, I\sigma^2)$ for $i = 1, \ldots, N$. Figure 8 shows the performance of the SSBM algorithm for different values of $\sigma$. The learning algorithm is particularly robust to noise, obtaining a good performance for noise levels up to $\sigma = 0.13$. This noise level translates to having approximately 15% of bits flipped per measurement vector. It is worth noting that these results indicate that we can expect similarly good performances for other noise distributions (*e.g.*, Poisson noise) for a similar average number of bit flips.

**Equivariant setting using shifts.** We evaluate the setting of learning with a single operator by using the unsupervised equivariant objective in (27) with 2D shifts as the group of transformations (as the MNIST dataset is approximately shift-invariant). Figure 9 shows the average test PSNR and reconstructed images as

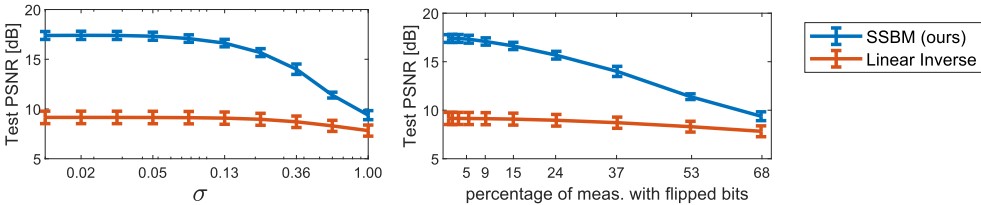

Figure 8: Robustness of the proposed learning algorithm to noise in the measurement data. **Left:** Average test PSNR as a function of the standard deviation of the noise. **Right:** Average test PSNR as a function of the average percentage of flipped bits per measurement vector.

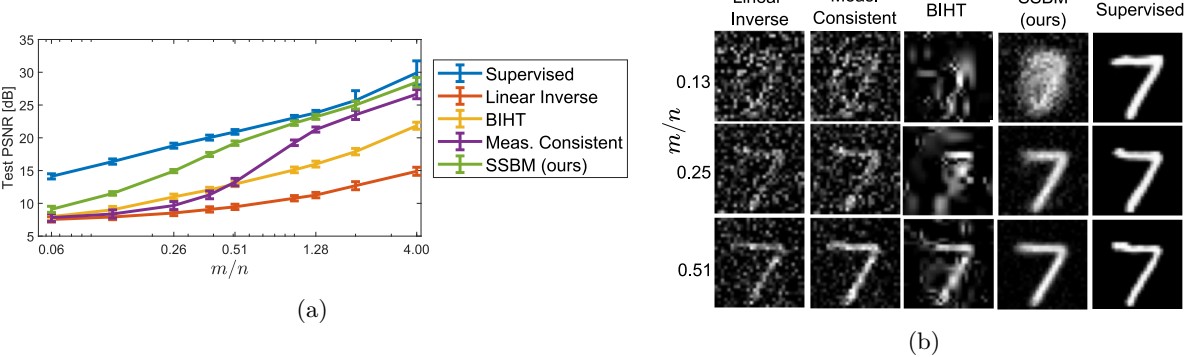

Figure 9: (**a**) Average test PSNR and (**b**) reconstructed test images by the compared algorithms with a single operator $A$ as a function of the undersampling ratio $m/n$.

a function of the measurements $m$ for various algorithms. The proposed unsupervised method significantly outperforms the linear inverse, BIHT, and the measurement consistent network in all sampling regimes, and performs closely to supervised learning for $m/n > 0.4$.

## 5.2 Other Datasets

In order to demonstrate the robustness of the proposed method across datasets, we evaluate the proposed unsupervised approach on the FashionMNIST (Xiao et al., 2017), CelebA (Liu et al., 2015) and Flowers (Nilsback & Zisserman, 2008) datasets. The FashionMNIST dataset consists of $6 \times 10^4$ greyscale images with $28 \times 28$ pixels which are divided across $G = 10$ different forward operators. As with MNIST, we use $N = 6 \times 10^3$ per operator for training and $10^3$ per operator for testing. For the CelebA dataset, we use $G = 10$ forward operators and choose a subset of $10^3$ images for each operator for training and another subset of the same amount for testing. The Flowers dataset consists of 6149 color images for training and 1020 images for testing, all associated with the same forward operator. For both CelebA and Flowers datasets, a center crop of $128 \times 128$ pixels of each color image was used for training and testing. Section 5.2 shows the average test PSNR of the proposed unsupervised method, standard supervised learning, BIHT, and the linear inverse. For BIHT, we use the Daubechies4 orthonormal wavelet basis and optimize the step size and sparsity level via grid search.

The self-supervised method obtains an average test PSNR which is only 1 to 2 dB below the supervised approach. Figures 10 and 11 show reconstructed test images by the evaluated approaches for each forward operator. The proposed unsupervised method is able to provide good estimates of the images, while only having access to highly incomplete binary information. The supervised method obtains sharper images, however at the cost of hallucinating details, whereas the proposed method obtains blurrier estimates with less hallucinated details.

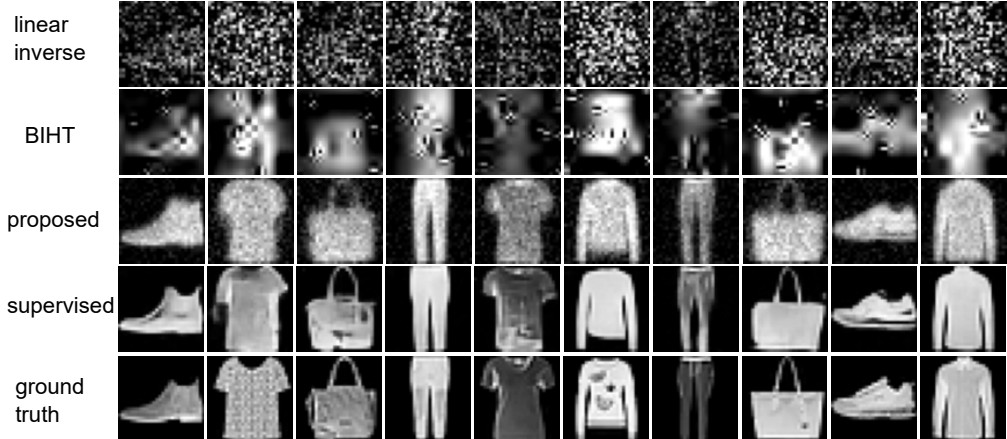

Figure 10: Reconstructed test images using the FashionMNIST dataset. Each column corresponds to a test image observed via a different forward operator $A_g$.

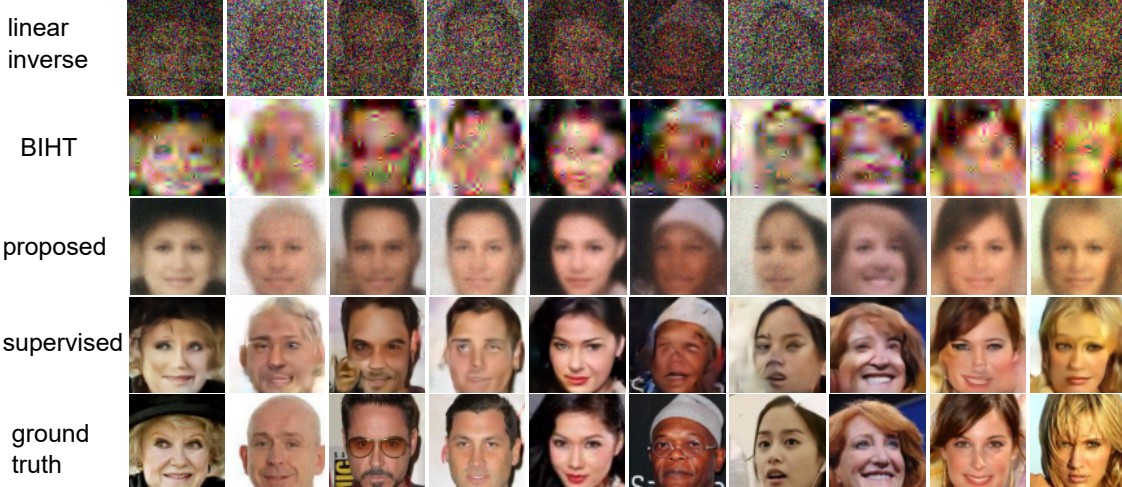

Figure 11: CelebA results. Reconstructed test images using the CelebA dataset. Each column corresponds to a test image observed via a different forward operator $A_g$.

## 6    Conclusions and Future Work

The theoretical analysis in this work characterizes the best approximation of a low-dimensional set that can be obtained from binary measurements. The model identification bounds presented here apply to a large class of signal models, as they only rely on the box-counting dimension, and complement those existing for signal recovery from binary measurements (Goyal et al., 1998; Jacques et al., 2013). Moreover, the proposed self-supervised loss provides a practical algorithm for learning to reconstruct signals from binary measurements alone, which performs closely to fully supervised learning. This work paves the way for deploying machine learning algorithms in scientific and medical imaging applications with quantized observations, where no ground-truth references are available for training.

We leave the proof of Conjecture 9, and a study of the effect of noise in the observations and related dithering techniques for future work. Another avenue of future research is the extension of Theorem 7 for the case of operators related through the action of a group.

| Dataset | $n$ | $m$ | $G$ | Linear Inverse | BIHT | Supervised | SSBM(ours) |
|---------|-----|-----|-----|----------------|------|------------|------------|
| FashionMNIST | 784 | 300 | 10 | $6.38 \pm 0.23$ | $10.68 \pm 0.31$ | $17.63 \pm 0.33$ | $16.47 \pm 0.22$ |
| CelebA | 49152 | 9830 | 10 | $4.81 \pm 0.32$ | $16.26 \pm 0.40$ | $21.59 \pm 0.31$ | $19.53 \pm 0.3$ |
| Flowers | 49152 | 9830 | shifts | $5.31 \pm 0.72$ | $14.62 \pm 0.92$ | $18.26 \pm 0.75$ | $16.45 \pm 0.71$ |

Table 2: Average test PSNR in dB obtained by the compared methods for the FashionMNIST, CelebA and Flowers datasets.

## Acknowledgments

Part of this research was supported by the Agence Nationale de la Recherche (Project UNLIP) and by the Fonds de la Recherche Scientifique – FNRS under Grant T.0136.20 (Project Learn2Sense).

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

# A Technical Lemmas

We begin by introducing some technical results that play an important role in the main theorems of the paper. We start with a result from (Jacques et al., 2013).

**Lemma 11** (Lemma 9 in (Jacques et al., 2013)). *Given* $0 \leq \epsilon < 1$ *and two unit vectors* $\tilde{x}, \tilde{v} \in \mathbb{S}^{n-1} \subset \mathbb{R}^n$ *and* $a \in \mathbb{R}^n$ *with* $a_i \sim_{\text{i.i.d.}} \mathcal{N}(0,1)$, *we have*

$$p_0 = \mathbb{P}\left[\forall x \in B_\epsilon(\tilde{x}), \forall v \in B_\epsilon(\tilde{v}) \ : \ \text{sign}\left(a^\top v\right) = \text{sign}\left(a^\top x\right)\right] \geq 1 - d(\tilde{x}, \tilde{v}) - \sqrt{n\frac{\pi}{2}}\epsilon \tag{35}$$

$$p_1 = \mathbb{P}\left[\forall x \in B_\epsilon(\tilde{x}), \forall v \in B_\epsilon(\tilde{v}) \ : \ \text{sign}\left(a^\top v\right) \neq \text{sign}\left(a^\top x\right)\right] \geq d(\tilde{x}, \tilde{v}) - \sqrt{n\frac{\pi}{2}}\epsilon. \tag{36}$$

*where* $d(\cdot, \cdot)$ *denotes the angular distance.*

**Remark:** The angular distances in Lemma 11 can be translated into Euclidean distances due to the following inequality:

$$d(\tilde{x}, \tilde{v}) \geq \frac{2}{\pi}\sin\left(\frac{\pi}{2}d(\tilde{x}, \tilde{v})\right) = \frac{1}{\pi}\|\tilde{x} - \tilde{v}\|. \tag{37}$$

Let $C^0(S)$ denote the set of continuous functions on the set $S$. This lemma has the following corollary:

**Corollary 12.** *Given* $\tilde{x} \in \mathbb{S}^{n-1}$, $0 < \epsilon < 1/2$, $a \in \mathbb{R}^n$ *with* $a \sim_{\text{i.i.d.}} \mathcal{N}(0,1)$, *we have*

$$\mathbb{P}\left[\text{sign}\left(a^\top \cdot\right) \notin C^0\left(B_\epsilon(\tilde{x}) \cap \mathbb{S}^{n-1}\right)\right] \leq \sqrt{n}\,\epsilon.$$

*Proof.* The proof can be derived from the complement of the event associated with $p_0$ in (35) when $\tilde{x} = \tilde{v}$. Here is, however, a simplified proof for completeness. We first observe that $\text{sign}\left(a^\top \cdot\right)$ is discontinuous over $B_\epsilon(\tilde{x}) \cap \mathbb{S}^{n-1}$ iff $\left|\frac{a^\top \tilde{x}}{\|a\|}\right| \leq \epsilon$. Therefore, by the rotational invariance of the Gaussian distribution we can choose $\tilde{x} = [1, 0, \ldots, 0]^\top$ and the probability above amounts to computing

$$p := \mathbb{P}[|\tfrac{a_1}{\|a\|}| \leq \epsilon] = \mathbb{P}[a_1^2 \leq \epsilon^2\|a\|^2] = \mathbb{P}[a_1^2 \leq \tfrac{\epsilon^2}{(1-\epsilon^2)}(a_2^2 + \ldots + a_n^2)] = \mathbb{E}_\xi\mathbb{P}[a_1^2 \leq \tfrac{\epsilon^2}{(1-\epsilon^2)}\xi],$$

where $\xi \sim \chi^2(n-1)$. Since $\mathbb{P}[a_1^2 \leq \tfrac{\epsilon^2}{(1-\epsilon^2)}\xi] \leq \frac{\sqrt{2}}{\sqrt{\pi}}\frac{\epsilon}{\sqrt{1-\epsilon^2}}\sqrt{\xi}$, and $\mathbb{E}_\xi\sqrt{\xi} \leq \sqrt{\mathbb{E}_\xi\xi} \leq \sqrt{n-1} \leq \sqrt{n}$ by Jensen's inequality, we finally get $p \leq \frac{\sqrt{2}}{\sqrt{\pi}}\frac{\epsilon}{\sqrt{1-\epsilon^2}}\sqrt{n} \leq \frac{2\sqrt{2}}{\sqrt{\pi}\sqrt{3}}\epsilon\sqrt{n} < \epsilon\sqrt{n}$. $\qquad\square$

# B Signal Recovery Proof

*Proof of Theorem 1.* Proving this theorem amounts to showing that the probability of the failure of the event

$$\text{sign}\left(Ax\right) = \text{sign}\left(As\right) \implies \|x - s\| < \delta$$

decays exponentially in $m$ provided that

$$m \geq \tfrac{4}{\delta}\left(2k \log \tfrac{30\sqrt{n}}{\delta} + \log \tfrac{1}{\xi}\right)$$

holds. In other words, we want to upper bound

$$p_\delta := \mathbb{P}\left[\exists x_1, x_2 \in \mathcal{X}, \|x_1 - x_2\| > \delta : \ \mathrm{sign}\left(Ax_1\right) = \mathrm{sign}\left(Ax_2\right)\right]$$

with such an exponential decay.

As $\mathrm{boxdim}\left(\mathcal{X}\right) < k$, there exist a constant $\epsilon_0 \in (0, \tfrac{1}{2})$ such that $\mathfrak{N}(\mathcal{X}, \epsilon) \leq \epsilon^{-k}$ for all $\epsilon \leq \epsilon_0$. Thus, there is a covering set $Q_\epsilon$ of $\epsilon^{-k}$ points, such that for every $x \in \mathcal{X}$, there exists a point $q \in Q_\epsilon$ which verifies $\|x - q\| < \epsilon$.

Thanks to this covering, we can upper bound $p_\delta$ as

$$p_\delta \ \leq \ \mathbb{P}\left[\exists q_1, q_2 \in Q_\epsilon, \exists x_1 \in B_\epsilon(q_1), \exists x_2 \in B_\epsilon(q_2), \|x_1 - x_2\| > \delta : \ \mathrm{sign}\left(Ax_1\right) = \mathrm{sign}\left(Ax_2\right)\right].$$

However, since $\|x_1 - x_2\| > \delta$, we must have $\|q_1 - q_2\| \geq \|x_1 - x_2\| - 2\epsilon > \delta - 2\epsilon$. Therefore, defining $Q_{\epsilon,\delta} = \{(q, q') \in Q_\epsilon \times Q_\epsilon : \|q - q'\| > \delta - 2\epsilon\}$, the previous upper bound can be enlarged as

$$p_\delta \ \leq \ \mathbb{P}\left[\exists (q_1, q_2) \in Q_{\epsilon,\delta}, \exists x_1 \in B_\epsilon(q_1), \exists x_2 \in B_\epsilon(q_2) : \ \mathrm{sign}\left(Ax_1\right) = \mathrm{sign}\left(Ax_2\right)\right].$$

Given a fixed pair $(q_1, q_2) \in Q_{\epsilon,\delta}$, Lemma 11 shows that for $a \in \mathbb{R}^n$ drawn from a standard Gaussian distribution

$$\mathbb{P}\left[\forall x_1 \in B_\epsilon(q_1), \forall x_2 \in B_\epsilon(q_2) : \ \mathrm{sign}\left(a^\top x_1\right) \neq \mathrm{sign}\left(a^\top x_2\right)\right] \geq \tfrac{1}{\pi}\|q_1 - q_2\| - \sqrt{\tfrac{\pi n}{2}}\epsilon > \tfrac{\delta - 2\epsilon}{\pi} - \sqrt{\tfrac{\pi n}{2}}\epsilon. \quad (38)$$

By setting $\epsilon = \epsilon(\delta) = \frac{\delta(4-\pi)}{8 + 4\pi\sqrt{n\pi/2}}$ and taking the probability of the complementary event we obtain

$$\mathbb{P}\left[\exists x_1 \in B_\epsilon(q_1), \exists x_2 \in B_\epsilon(q_2) : \ \mathrm{sign}\left(a^\top x_1\right) = \mathrm{sign}\left(a^\top x_2\right)\right] \leq 1 - \delta/4. \quad (39)$$

Therefore, considering the $m$ i.i.d. rows $\{a_i\}_{i=1}^m \subset \mathbb{R}^n$ of the matrix $A = (a_1, \ldots, a_m)^\top \in \mathbb{R}^{m \times n}$ drawn from a standard Gaussian distribution, we have

$$\begin{aligned}
&\mathbb{P}[\exists x_1 \in B_\epsilon(q_1), \exists x_2 \in B_\epsilon(q_2) : \ \mathrm{sign}\left(Ax_1\right) = \mathrm{sign}\left(Ax_2\right)] \\
&\leq \ \textstyle\prod_{i=1}^m \mathbb{P}[\exists x_1 \in B_\epsilon(q_1), \exists x_2 \in B_\epsilon(q_2) : \ \mathrm{sign}\left(a_i^\top x_1\right) = \mathrm{sign}\left(a_i^\top x_2\right)] \\
&\leq \ (1 - \delta/4)^m.
\end{aligned}$$

Applying a union bound to all pairs $(q_1, q_2) \in Q_{\epsilon(\delta),\delta} \subset Q_\epsilon \times Q_\epsilon$, since there are no more that $\binom{|Q_\epsilon|}{2} \leq |Q_\epsilon|^2 \leq \epsilon^{-2k}$ such pairs, we obtain

$$p_\delta \leq \left(\tfrac{8 + 4\pi\sqrt{\pi n/2}}{(4-\pi)\delta}\right)^{2k}(1 - \delta/4)^m \leq \exp\left(2k \log\left(\tfrac{8 + 4\pi\sqrt{\pi n/2}}{(4-\pi)\delta}\right) - \tfrac{m\delta}{4}\right), \quad (40)$$

where we used $1 - \delta/4 \leq \exp(-\delta/4)$ for $\delta > 0$.

Upper bounding this probability by $0 \leq \xi \leq 1$ as in the statement of Thm 1 and using the crude bound $(8 + 4\pi\sqrt{\pi n/2})/(4 - \pi) \leq 30\sqrt{n}$ for $n \geq 1$, we finally obtain $2k \log \tfrac{30\sqrt{n}}{\delta} + m\tfrac{\delta}{4} \geq \log \xi$ which gives the sample complexity bound (9)

$$m \geq \tfrac{4}{\delta}\left(2k \log \tfrac{30\sqrt{n}}{\delta} + \log \tfrac{1}{\xi}\right), \quad (41)$$

where the condition $\epsilon(\delta) \leq \epsilon_0$ holds if $\delta \leq 30\epsilon_0\sqrt{n}$.

$\square$

## C   Model Identification Proof

*Proof of Theorem 7.* We want to identify the condition that $m$, $G$ and $0 < \delta < 1$ must respect to induce that $\hat{\mathcal{X}} \subseteq \mathcal{X}_\delta$ holds with high probability with respect to a random draw of the operators $A_1, \ldots, A_G$. Equivalently, we need to show that, for this condition,

$$\operatorname{sign}(A_g x_g) = \operatorname{sign}(A_g v), \quad \forall g = 1, \ldots, G \tag{42}$$

holds for some $v \in \mathbb{S}^{n-1} \setminus \mathcal{X}_\delta$ and some $x_1, \ldots, x_G \in \mathcal{X}$ with probability at most $\xi$ with respect to a random draw of the Gaussian matrices $A_1, \ldots, A_G$. This proof adapts some of the procedures given in (Jacques et al., 2013) to our specific setting. We start by bounding this probability for $\epsilon$-balls around vectors $\tilde{v} \in \mathbb{S}^{n-1} \setminus \mathcal{X}_\delta$, $\tilde{x}_1, \ldots, \tilde{x}_G \in \mathcal{X}$, that is

$$p_0 := \mathbb{P}\big[\exists (x_1, \ldots, x_G) \in B_\epsilon(\tilde{x}_1) \times \cdots \times B_\epsilon(\tilde{x}_G), \exists v \in B_\epsilon(\tilde{v}) : \forall g = 1, \ldots, G, \ \operatorname{sign}(A_g v) = \operatorname{sign}(A_g x_g)\big].$$

We first notice that from the independence of the operators $\{A_g\}_{g=1}^G$,

$$p_0 \le \prod_{g=1}^G \mathbb{P}\big[\exists x_g \in B_\epsilon(\tilde{x}_g), \exists v \in B_\epsilon(\tilde{v}) : \operatorname{sign}(A_g v) = \operatorname{sign}(A_g x_g)\big].$$

Furthermore, as every row of each operator $A_g$ is i.i.d. as a standard Gaussian random vector $a_g$, we have

$$p_0 \le \prod_{g=1}^G \mathbb{P}\big[\exists x_g \in B_\epsilon(\tilde{x}_g), \exists v \in B_\epsilon(\tilde{v}) : \operatorname{sign}(a_g^\top v) = \operatorname{sign}(a_g^\top x_g)\big]^m \tag{43}$$

$$= \prod_{g=1}^G \Big(1 - \mathbb{P}\big[\forall x_g \in B_\epsilon(\tilde{x}_g), \forall v \in B_\epsilon(\tilde{v}) : \operatorname{sign}(a_g^\top v) \ne \operatorname{sign}(a_{g,i}^\top x_g)\big]\Big)^m \tag{44}$$

From Lemma 11, we know that

$$\mathbb{P}\big[\forall x_g \in B_\epsilon(\tilde{x}_g), \forall v \in B_\epsilon(\tilde{v}) : \operatorname{sign}(a_{g,i}^\top v) \ne \operatorname{sign}(a_{g,i}^\top x_g)\big] \ge \tfrac{1}{\pi}\|\tilde{x}_g - \tilde{v}\| - \sqrt{n\tfrac{\pi}{2}}\epsilon. \tag{45}$$

where the distance $\|\tilde{x}_g - \tilde{v}\|$ can be bounded by $\delta$ to obtain

$$\mathbb{P}\big[\forall x_g \in B_\epsilon(\tilde{x}_g), \forall v \in B_\epsilon(\tilde{v}) : \operatorname{sign}(a_{g,i}^\top v) \ne \operatorname{sign}(a_{g,i}^\top x_g)\big] \ge \tfrac{\delta}{\pi} - \sqrt{n\tfrac{\pi}{2}}\epsilon. \tag{46}$$

Plugging this into (44) and picking $\tfrac{\delta}{\pi} - \sqrt{n\tfrac{\pi}{2}}\epsilon = \tfrac{\delta}{4}$, which means that

$$\epsilon = \epsilon(\delta) = \big(\tfrac{4-\pi}{\sqrt{8\pi^3}}\big)\tfrac{\delta}{\sqrt{n}} \le \tfrac{1}{18}\tfrac{\delta}{\sqrt{n}},$$

we get

$$p_0 \le \big(1 - \tfrac{\delta}{\pi} + \sqrt{n\tfrac{\pi}{2}}\epsilon\big)^{mG} \le \exp(-\tfrac{\delta}{4}mG). \tag{47}$$

We can extend this result to all vectors $v \in \mathbb{S}^{n-1} \setminus \mathcal{X}_\delta$ and $x_1, \ldots, x_G \in \mathcal{X}$ by applying a union bound over a covering of the product set $\mathcal{X}^G \times (\mathbb{S}^{n-1} \setminus \mathcal{X}_\delta)$. Since we can cover $\mathcal{X}$ with $\epsilon^{-k}$ balls with $\epsilon \le \epsilon_0$ due to the assumption that $\operatorname{boxdim}(\mathcal{X}) < k$, and also cover $\mathbb{S}^{n-1} \setminus \mathcal{X}_\delta$ with $(3/\epsilon)^n$ balls (Pisier, 1999), we have

$$\mathbb{P}[\exists x_1, \ldots, x_G \in \mathcal{X}, \exists v \in (\mathbb{S}^{n-1} \setminus \mathcal{X}_\delta) : \forall g = 1, \ldots, G, \ \operatorname{sign}(A_g v) = \operatorname{sign}(A_g x_g)] \le \epsilon^{-kG}(\epsilon/3)^{-n}p_0 \tag{48}$$

Using the bound (47), the upper bound on $\epsilon$, and bounding the resulting probability by $\xi$, we obtain

$$\xi \ge \epsilon^{-kG}\big(\tfrac{\epsilon}{3}\big)^{-n}\exp(-\tfrac{\delta}{4}mG) = \exp\big(kG\log(\tfrac{1}{\epsilon}) + n\log(\tfrac{3}{\epsilon}) - \tfrac{\delta}{4}mG\big)$$

$$\ge \exp\big[kG\log(\tfrac{18\sqrt{n}}{\delta}) + n\log(\tfrac{54\sqrt{n}}{\delta}) - \tfrac{\delta}{4}mG\big].$$

Equivalently, $m \ge \tfrac{4}{\delta}\big[k\log(\tfrac{18\sqrt{n}}{\delta}) + \tfrac{n}{G}\log(\tfrac{54\sqrt{n}}{\delta}) + \tfrac{1}{G}\log(1/\xi)\big]$, which holds if

$$m \ge \tfrac{4}{\delta}\big[(k + \tfrac{n}{G})\log(\tfrac{54\sqrt{n}}{\delta}) + \tfrac{1}{G}\log(\tfrac{1}{\xi})\big].$$

Recalling, we must have $\epsilon < \epsilon_0$, we observe that this conditions is met if $\delta < 18\sqrt{n}\,\epsilon_0$.

$$\square$$

**Derivation of $\delta$.** Here we aim to upper bound the minimum identification error, *i.e.*, the minimum value of $\delta$, for a fixed number of measurements $m$. The bound in Theorem 7, that is

$$m \geq \tfrac{4}{\delta}\big((k + \tfrac{n}{G})\log(\tfrac{54\sqrt{n}}{\delta}) + \tfrac{1}{G}\log(\tfrac{1}{\xi})\big) \tag{49}$$

can be rewritten as a function of $\delta$ as

$$\log(\delta) + \delta a \geq b \tag{50}$$

where

$$a = \tfrac{m}{4(k + \frac{n}{G})}, \quad \text{and} \quad b = \log 54\sqrt{n} + \tfrac{1}{(Gk+n)}\log\tfrac{1}{\xi}.$$

Notice that $b \geq 1$, and $a \geq 1$ from (49) since $0 < \delta < 1$. The expression in (50) holds if

$$\delta \geq \tfrac{1}{a}(\log(a) + b). \tag{51}$$

Indeed, (51) implies that $\log(\delta) + \delta a \geq \log(\delta) + \log(a) + b = \log(a\delta) + b$. However, again from (51), we get $a\delta \geq \log(a) + b \geq 1$ since $a, b \geq 1$. Therefore, $\log(\delta) + \delta a \geq \log(a\delta) + b \geq b$.

Finally, picking the smallest $\delta$ respecting (51), we get for large $m$, $n$ and $G$,

$$\delta = \mathcal{O}\big(\tfrac{k + \frac{n}{G}}{m}\log\tfrac{mn}{k + \frac{n}{G}}\big) \tag{52}$$

which, for $n/G \ll k$ reads

$$\delta = \mathcal{O}\big(\tfrac{k}{m}\log\tfrac{mn}{k}\big). \tag{53}$$

# D  Sample Complexity Proof

*Proof of Theorem 10.* We aim to bound the number of different cells associated with the binary mapping $\mathrm{sign}\,(A\cdot)$ which contain at least one element from the signal set $\mathcal{X}$, *i.e.*, $|\,\mathrm{sign}\,(A\mathcal{X})\,|$. Our strategy consists in obtaining a global bound on the number of discontinuities of the binary mapping (or in other words, sign changes) over the image of a covering of $\mathcal{X}$, which can then be related to the number of different cells that contain at least one element of $\mathcal{X}$.

For $\epsilon < \epsilon_0$, let $Q_\epsilon \subset \mathcal{X}$ be an optimal $\epsilon$ covering of $\mathcal{X}$. If $\mathrm{boxdim}\,(\mathcal{X}) < k$, then there exists an $\epsilon_0 \in (0, \tfrac{1}{2})$ such that $|Q_\epsilon| \leq \epsilon^{-k}$ for all $\epsilon < \epsilon_0$. Let us define the number $Z(S)$ of its discontinuous components of the binary mapping $\mathrm{sign}\,(A\cdot)$ over a set $S \subset \mathbb{S}^{n-1}$, *i.e.*,

$$Z(S) := \big|\{i : \mathrm{sign}\,(a_i^\top \cdot) \notin C^0(S)\}\big|.$$

From the independence of the $\{a_i\}_{i=1}^m$, we observe that $Z(S) = \sum_{i=1}^m Z_i(S)$ is a binomial random variable, that is the sum of $m$ i.i.d. binary random variables with probability

$$p := \mathbb{P}\Big[\mathrm{sign}\,(a^\top \cdot) \notin C^0(S)\Big],$$

where $a$ is a standard Gaussian random vector. From Corollary 12, we have $p \leq \sqrt{n}\epsilon$ for any set of the form $S = S_{q,\epsilon} := B_\epsilon(q) \cap \mathbb{S}^{n-1}$ with $q \in \mathbb{R}^n$. Using Bernstein inequality (Vershynin, 2018) on the random variable $Z(S_{q,\epsilon})$ with $\mathbb{E}Z(S_{q,\epsilon}) = mp \leq m\sqrt{n}\epsilon$, we obtain

$$\mathbb{P}[Z(S_{q,\epsilon}) > 2m\sqrt{n}\epsilon] \leq \mathbb{P}[Z > 2\mathbb{E}Z] \leq \exp(-\tfrac{3}{8}m\sqrt{n}\epsilon).$$

Therefore, since $|Q_\epsilon| \leq \epsilon^{-k}$, we get from a union bound

$$\mathbb{P}[\forall q \in Q_\epsilon : \ Z(S_{q,\epsilon}) \leq 2m\sqrt{n}\epsilon] \leq 1 - \exp(k\log(\tfrac{1}{\epsilon}) - \tfrac{3}{8}m\sqrt{n}\epsilon). \tag{54}$$

Let us fix $\epsilon$ by setting a failure probability $0 < \xi < 1$ such that $\xi = \exp(k\log(\tfrac{1}{\epsilon}) - \tfrac{3}{8}m\sqrt{n}\epsilon)$, *i.e.*,

$$2m\sqrt{n}\epsilon = \tfrac{16}{3}\big[\log(\tfrac{1}{\xi}) + k\log(\tfrac{1}{\epsilon})\big]. \tag{55}$$

This implicitly imposes $\frac{3}{8}m\sqrt{n}\epsilon > k\log(\frac{1}{\epsilon})$, and since $\epsilon < \min(\epsilon_0, 1/2)$, we get $\frac{3}{8}m\sqrt{n}\epsilon > k\log 2$, or

$$\frac{8k\log 2}{3m\sqrt{n}} < \epsilon. \tag{56}$$

Since the left-hand side of (56) has to be smaller than $\min(\epsilon_0, 1/2)$, we have that $2k/(m\sqrt{n}) < \min(\epsilon_0, 1/2)$ (using the fact that $2 > \frac{8}{3}\log 2$).

For any set $S \subseteq \mathbb{S}^{n-1}$, the number of cells generated by $\text{sign}(A\cdot)$ in this set cannot exceed 2 to the power of the number of discontinuous components in this mapping, $i.e.$, $|\text{sign}(AS)| \leq 2^{Z(S)}$. Thus, with probability $1 - \xi$, and given $q \in Q_\epsilon$,

$$|\text{sign}(AS_{q,\epsilon})| \leq 2^{\frac{16}{3}[\log(\frac{1}{\xi}) + k\log(\frac{1}{\epsilon})]} = \left(\frac{1}{\xi}\right)^{\frac{16\log 2}{3}}\left(\frac{1}{\epsilon}\right)^{\frac{16\log 2}{3}k} < \left(\frac{1}{\xi}\right)^4\left(\frac{1}{\epsilon}\right)^{4k}.$$

Since there are at most $\epsilon^{-k}$ balls in the covering, and using (56), we obtain the bound

$$|\text{sign}(A\mathcal{X})| \leq \sum_{q \in \mathcal{X}_\epsilon} |\text{sign}(AS_{q,\epsilon})| < \left(\frac{1}{\xi}\right)^4\left(\frac{1}{\epsilon}\right)^{5k} < \left(\frac{1}{\xi}\right)^4\left(\frac{3m\sqrt{n}}{8k\log 2}\right)^{5k} < \left(\frac{1}{\xi}\right)^4\left(\frac{3m\sqrt{n}}{5k}\right)^{5k}.$$

We now prove a bound on the expected number of intersected cells. We first observe that

$$\mathbb{E}|\text{sign}(A\mathcal{X})| \leq \sum_{q \in \mathcal{X}_\epsilon} \mathbb{E}|\text{sign}(AS_{q,\epsilon})|,$$

and, for any set $S \subseteq \mathbb{S}^{n-1}$, the independence of the random variables $Z_i(S)$ provides

$$\mathbb{E}|\text{sign}(AS)| \leq \mathbb{E}\,2^{Z(S)} = \mathbb{E}\big[2^{\sum_{i=1}^m Z_i(S)}\big] = \mathbb{E}\big[\prod_i 2^{Z_i(S)}\big] = \prod_i \mathbb{E}\,2^{Z_i(S)}.$$

Moreover, considering the previous covering $\mathcal{Q}_\epsilon$ of $\mathcal{X}$, if $S = S_{q,\epsilon}$, for some $q \in \mathcal{Q}_\epsilon$,

$$\mathbb{E}2^{Z_i(S)} = 2^0\mathbb{P}(Z_i(S) = 0) + 2^1\mathbb{P}(Z_i(S) = 1) = (1 - p) + 2p = 1 + p \leq 1 + \sqrt{n}\epsilon \leq e^{\sqrt{n}\epsilon}.$$

Therefore, $\mathbb{E}|\text{sign}(AS_{q,\epsilon})| \leq e^{m\sqrt{n}\epsilon}$, and

$$\mathbb{E}|\text{sign}(A\mathcal{X})| \leq \epsilon^{-k}e^{m\sqrt{n}\epsilon} = e^{k\log(\frac{1}{\epsilon}) + m\sqrt{n}\epsilon}.$$

The function $k\log(\frac{1}{\epsilon}) + m\sqrt{n}\epsilon$ is convex in $\epsilon$ and reaches its minimum on $\epsilon = \frac{k}{m\sqrt{n}}$. Therefore, by setting $\epsilon = k/(m\sqrt{n})$ and imposing $k/m\sqrt{n} \leq \min(\epsilon_0, 1/2)$, we get

$$k\log(\tfrac{m\sqrt{n}}{k}) + k = k(1 + \log(\tfrac{m\sqrt{n}}{k})),$$

which finally gives

$$\mathbb{E}|\text{sign}(A\mathcal{X})| \leq e^{k(1 + \log(\frac{m\sqrt{n}}{k}))} = (e\tfrac{m\sqrt{n}}{k})^k.$$

$\square$

# E  Choice of trade-off parameter

We evaluate the impact of the trade-off parameter of the proposed SSBM algorithm for different sampling ratios $m/n$ on the MNIST dataset. In all cases, we use $G = 10$ operators. Figure 12 shows the average test PSNR of the learned networks. The optimal choice of $\alpha$ decreases with the number of measurements $m$. In the experiments, we choose $\alpha = 0.1$ if $m < n$ and $\alpha = 0.06$ otherwise.

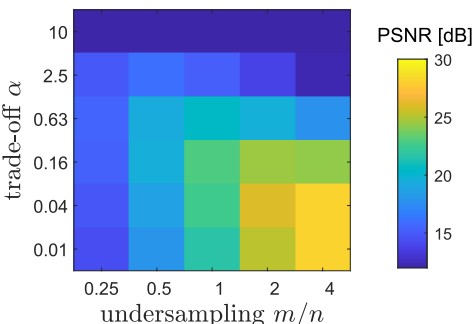

Figure 12: Impact of the trade-off parameter $\alpha$ of the SSBM learning algorithm as a function of the sampling ratio $m/n$ network for the MNIST problem with $G = 10$ operators.

