# OpenReview forum: "Learning to reconstruct signals from binary measurements alone"
_TMLR — Accepted by TMLR_

### Review · Reviewer_gnaz · 2023-08-01

**Summary Of Contributions:**

This paper develops, analyzes, and tests an unsupervised learning (learn using only noisy observations) algorithm for reconstructing signals from binary linear measurements (1-bit compressive sensing). The first few sections theoretically lower bound the reconstruction error as a function of the number of measurement matrix size and the number of observations.  The last few sections develop a simple algorithm (essentially equivariant imaging) to train a network to reconstruct signals from binary measurements. It's tested and is proven reasonably effective on MNIST, Fashion MNIST, on CelebA datasets.


**Audience:**

Yes

**Claims And Evidence:**

Yes

**Requested Changes:**

On the first page, I'd use a different word than "incomplete" (maybe "underdetermined" or "fat") to describe the measurement matrix. The word "incomplete" has too many alternative meanings, e.g., matrix completion. I'd similarly avoid using "incomplete" to describe the measurement process.

I think equation (10) should be an equality between the measurements. Presently (10) states that if the measurements of two points disagree then they must be close to one another with some probability.

The notation in (14) seems slightly odd. Would it be clearer to write X_d={v\in S^{n-1}| inf_{x\in X} ||x-v||<d}

I think the paper would be stronger if one of the tests (e.g., Fig 9) were rerun with varying amounts of measurement noise. At present, it's unclear how sensitive the proposed method is to noise.

Typos:
At least on my pdf reader, the Fashion MNIST and MNIST dataset descriptions are missing multiplication signs: 6 10^5


**Strengths And Weaknesses:**

## Strengths
Clarity: Paper is well-written and figures like 2 are intuitive and useful.

Novelty: To my knowledge unsupervised learning methods like equivariant imaging have not been applied to binary measurements.


## Weaknesses
The theory/results are developed/tested without any measurement noise.

---

> ### Author Response · Authors · 2023-09-27
> **Response to reviewer gnaz**
>
> Thank you for the comments about our work. We have uploaded a revised manuscript with changes highlighted in red.
>
> > The theory/results are developed/tested without any measurement noise.
>
> In the setting of binary measurements, noise affects the measurements by flipping the sign, as the observations can only be $-1$ or $+1$. Thus, the proposed method is particularly robust to noise, as it is only affected by noise that is large enough to produce bit flips in the measurement vector. We have included new experiments demonstrating the robustness of the proposed method to noise (see Figure 8 of the revised manuscript). We have included this discussion on the impact of noise in the updated manuscript (see page 15).
>
> Please see more details in the answer to one of the points below.
>
> > On the first page, I'd use a different word than "incomplete" (maybe "underdetermined" or "fat") to describe the measurement matrix. The word "incomplete" has too many alternative meanings, e.g., matrix completion. I'd similarly avoid using "incomplete" to describe the measurement process.
>
> Thank you for this suggestion. It is worth mentioning that the link with matrix completion is not wrong, since our method can be seen as a one-bit matrix completion approach where the columns of the matrix are assumed to belong to a distribution with low-dimensional support, going beyond a simple linear subspace (which is the assumption behind standard low-rank matrix completion). This point is discussed in the *Related Work* subsection of the manuscript.
>
> We prefer not to use the word "fat" or "underdetermined" as they refer specifically to the properties of the matrix, and do not extrapolate well to the sensing process. For example, while an *incomplete measurement process* is meaningful, a *fat measurement process* is not. We have also avoided the use of *compressed* since it is often wrongly associated with random matrices, whereas the necessary conditions presented in this paper are also valid for deterministic matrices. Given these points, we prefer to keep the current naming.
>
> > I think equation (10) should be an equality between the measurements. Presently (10) states that if the measurements of two points disagree then they must be close to one another with some probability.
>
> We agree with the reviewer, this was an undesired typo in equation (10). We have corrected it in the updated manuscript to the equality $\text{sign}(Ax)=\text{sign}(Au)$.
>
> > The notation in (14) seems slightly odd. Would it be clearer to write $\\mathcal{X}\_{\\delta}=\\{v\\in \\mathbb{S}\^{n-1}\| \\inf\_{x\\in
> \\mathcal{X}} \\\|x-v\\\|\<{\\delta}\\}\$.
>
> Thank you for this suggestion that we have implemented.
>
> > I think the paper would be stronger if one of the tests (e.g., Fig. 9) were rerun with varying amounts of measurement noise. At present, it's unclear how sensitive the proposed method is to noise.
>
> We have included a new experiment evaluating the proposed algorithm with $m=274$ measurements and $G=10$ operators and different noise levels, according to the model
> \begin{equation}
>     y_i = \text{sign}\left(A_{g_i,i}x_i + \epsilon_i \right)
> \end{equation}
> where $\epsilon_i\sim \mathcal{N}(0,I\sigma^2)$ for $i=1,\dots,N$. Figure 8 of the updated manuscript shows the performance of the SSBM algorithm for different values of $\sigma$. The learning algorithm is particularly robust to noise, obtaining a good performance for noise levels up to $\sigma=0.13$. This noise level translates to having approximately 15\% of bits flipped per measurement vector. It is worth noting that these results indicate that we can expect similarly good performances for other noise distributions (e.g., Poisson noise) for a similar average number of bit flips.
>
> > Typos: At least on my pdf reader, the Fashion MNIST and MNIST dataset descriptions are missing multiplication signs: $6\times10^5$
>
> Thanks for spotting these typos, we have updated them in the new manuscript to $6\times 10^4$ samples ($6\times 10^3$ samples per operator, for $G=10$ operators) for both MNIST and FashionMNIST (there was a typo in the case of FashionMNIST in the previous version, wrongly saying it has $6\times 10^5$ samples).

---

### Review · Reviewer_x5u4 · 2023-08-29

**Summary Of Contributions:**

This paper considers the problem of one-bit compressed sensing, where we observe a one-bit quantization $y$ of the signal $x$ after it went through a sensing matrix $A$. While previous works focused on the reconstruction of $x$ given the knowledge of the signal set $\mathcal X$, this work focuses on reconstructing the signal set $\mathcal X$ itself based on multiple observations $y_i$.

The paper provides upper and lower bounds for this reconstruction set, that match up to logarithmic factors if the set $\mathcal X$ has low box-counting dimension. They also extend previous results on signal reconstruction from the sparse setting to the more general case with low box-counting dimensions.

Finally, the authors propose a loss for self-supervised learning of the signal, and provide a set of experiments to compare it with supervised or other non-supervised methods.

**Audience:**

Yes

**Claims And Evidence:**

Yes

**Requested Changes:**

All of the changes can be considered as non-critical

- The identification error should be actually defined, e.g. as $\min \\{\delta: \hat{\mathcal X} \subseteq \mathcal X_\delta\\}$

- It might be useful to add a sentence or two after Proposition 2, to explain exactly what has been proven

- Proof of Prop.3:
    - $v$ is not a necessarily a generator of the nullspace of $\bar A$, only an element of it
    - it'd be more intuitive to specify directly that $x$ is in $\mathrm{Ker}(\bar A)^\bot$, instead of the range of $\bar A^\top$ (even though it's equivalent)
    - I personally would've found it more intuitive to fix $x \in \mathbb S^{n-1}$ and consider $x \pm \eta v$ where $\eta \to \infty$, but this is a personal preference

- Experiments:
    - you mention the "necessary condition of Proposition 3" multiple times; you should either move the $m > n/G$ inside this proposition, or (my preferred option) put this condition in a corollary and refer to it instead.

- Proof of Theorem 1:
    - the link from (34) to (35) could use a (simple) drawing to explain what's going on
    - what does the $|$ sign mean in the probabilities ? In any case, I think it should be replaced by "and"

- Derivation of $\delta$:
    - the link from (62) to (63) is unclear, but you can get (63) by the simpler inequality $W(x) > \ln(x)/2$ valid for any $x \in \mathbb R$

- Proof of Theorem 8: I feel like the use of discontinuity points is a bit circular and non-intuitive; the argument in the proof can be summarized as
$$ \text{number of cells} \asymp \text{number of cell borders} \leq \sum_{q \in Q_\epsilon} \text{number of borders in } B(q, \epsilon)$$
which is more intuitive than the discontinuity argument.

**Strengths And Weaknesses:**

This is overall a very good paper. The results proven are novel, relevant to both theoretical and practicioner audiences interested in self-supervised learning in compressed sensing, and rigorously proven. Even with low prior knowledge about this topic, the paper is easy to follow, with helpful diagrams explaining the key concepts of binary recovery.

It does have some unclear parts in the proofs (see the requested changes section), but this paper is overall very well suited for a TMLR publication.

---

> ### Author Response · Authors · 2023-09-27
> **Response to reviewer x5u4**
>
> We thank the reviewer for the interesting and detailed feedback. We have uploaded a revised manuscript with changes highlighted in red.
>
> > The identification error should be actually defined, e.g. as $ \\min \\{\delta: \mathcal{X} \subseteq \mathcal{X}_\delta \\} $
>
> Indeed, thank you. We have included a formal definition of the identification error in Section 3 of the updated manuscript (see Definition 3.1).
>
> > It might be useful to add a sentence or two after Proposition 2, to explain exactly what has been proven.
>
> In words, Proposition 2 shows that we can rotate any signal set $\mathcal{X}$ such that it intersects the largest consistency cell in the tesselation, obtaining a model identification error that is proportional to the maximum cell diameter. The rotation is used to remove the best-case scenarios where the signal set only intersects consistency cells that are smaller than the largest one. We have included this discussion after the proof of Proposition 2 in the updated manuscript.
>
> > Proof of Prop.3: is not necessarily a generator of the nullspace of $\bar{A}$, only an element of it
>
> We have changed the word *generator* for *element* in the updated manuscript, to clarify that the vector $v$ does not generate the whole nullspace of $\bar{A}$.
>
> > Proof of Prop.3: It'd be more intuitive to specify directly that $x$ is in the complement of $\text{Ker}(\bar{A})^\perp$, instead of the range of $\bar{A}^{\top}$ (even though it's equivalent)
>
> Thank you for this suggestion. We have inserted this equivalent formulation.
>
> >  Proof of Prop.3: I personally would've found it more intuitive to fix $x\in \mathbb{S}^{n-1}$ and consider $x \eta$ where $\eta\to\infty$, but this is a personal preference.
>
> We agree with this interesting approach, however, we have decided to keep the proposed solution.
>
> > Experiments: You mention the "necessary condition of Proposition 3" multiple times; you should either move the $m>n/G$ inside this proposition, or (my preferred option) put this condition in a corollary and refer to it instead.
>
> We have included a corollary to make this condition more clear as suggested by the reviewer (see Corollary 4 of the updated manuscript). The references appearing later on in the text refer now directly to the corollary.
>
> > Proof of Theorem 1: The link from (34) to (35) could use a (simple) drawing to explain what's going on. What does the $|$ sign mean in the probabilities? In any case, I think it should be replaced by "and".
>
> Following the comments of reviewer DqqG, the proof of this theorem has been updated and corrected, stating from the beginning the pursued objective and how Lemma 9 (now Lemma 11) is used. Moreover, this rewriting also avoids the use of a *condition* sign "$|$" altogether. We believe the explanations are clearer now.
>
> > Derivation of $\delta$: the link from (62) to (63) is unclear, but you can get (63) by the simpler inequality $W(x)>\log x/2$ valid for any $x\in \mathbb{R}$.
>
> We have simplified the derivation of $\delta$ in the updated manuscript  (see page 22 of the updated manuscript), avoiding altogether the use of the Lambert function $W(x)$. The new derivation should solve the reviewer's comment, while (in our opinion) being easier to follow.
>
> > Proof of Theorem 8: I feel like the use of discontinuity points is a bit circular and non-intuitive; the argument in the proof can be summarized as
> $$
>     \text{number of cells} \asymp \text{number of cell borders} \leq \sum_{q\in Q_{\epsilon}} \text{number of borders in $B(q,\epsilon)$}
> $$
> which is more intuitive than the discontinuity argument.
>
> We have modified the proof of Theorem 8 (Theorem 10 in the revised manuscript) to render it more clear to the reader. The updated proof contains an explicit explanation of the link between the number of discontinuities and the number of intersected cells, which was missing in the first submission and may have caused confusion. In particular, the number of discontinuities $Z(S)$ of the binary mapping can be used to upper bound the number of intersected cells $|\text{sign}(AS)|$, by noting that $|\text{sign}(AS)|\leq 2^{Z(S)}$ for any set $S\subseteq \mathbb{S}^{n-1}$. For completeness, we've also added an upper bound on the expectation $\mathbb E|\text{sign}(A\mathcal{X})|$ that scales similarly to the probabilistic bound.

---

### Review · Reviewer_DqqG · 2023-09-06

**Summary Of Contributions:**

This paper focuses on the problem of reconstructing a set $\mathcal{X}$ from binary measurements of the type $y_i = \mathrm{sign}(A x_i)$ for $x \in \mathcal{X}$.
The authors tackle two problems: signal reconstruction (i.e.\ estimating the inverse mapping $y \mapsto x$) and model identification, i.e. the identification of the set $\mathcal{X}$ provided this set is effectively low-dimensional.
The setup chosen is challenging, as the authors aim for a study of unsupervised learning procedures, in which the statistician only has access to $(y_i)_{i=1}^N$, and not the ground-truth signals $(x_i)$.

To make it tracktable, two separate setups are assumed:

* The measurement operator varies accross observations, that is $y_i = \mathrm{sign}(A_{g_i} x_i) \in \mathbb{R}^m$, and $g_i \in [G]$. There are here $G$ different (known) measurement operators.
* The measurement operator does not vary, but the set $\mathcal{X}$ is known to be invariant under the representation $(T_g)_{g \in [G]}$ of a finite group.

In both cases above, the authors derive a set of three theoretical results:

1. A worst-case bound w.r.t. $(A_g)_{g \in [G]}$ in Section 3.1. For any such measurement operators, there exists a set $\mathcal{X} \subseteq \mathcal{S}^{n-1}$ such that one can not infer $\mathcal{X}$ with an error smaller than $\mathcal{O}(n / (mG))$. Having error at least $\delta$ here means that there are points in the estimated set $\hat{\mathcal{X}}$ which are at distance at least $\delta$ from the true $\mathcal{X}$.
2. An upper bound on the estimation error if $A_g$ are taken i.i.d. from a standard Gaussian distribution, in Section 3.2. If $\mathcal{X}$ has ``effective dimension'' $k$, then the error is (with high probability) at most $\mathcal{O}([(k+n/G)/n] \log[nm/(k+n/G)])$, which is sharp up to the logarithmic factor. Furthermore, the authors conjecture (in Section 3.4) that the same error estimate should hold for the signal reconstruction procedure.
3. Again in the case of Gaussian measurement operators, the authors show in Section 3.4 an upper bound on the minimal number of distinct measurements needed to perform an optimal estimation of the set $\mathcal{X}$, as a function of $m, n, G$ and the effective dimension $k$ of $\mathcal{X}$.

The proofs of the main results are based on classical tools of high-dimensional probability (covering arguments and concentration inequalities). Sections 4 and 5 are dedicated to introducing a self-supervised procedure (i.e. that does not need access to the ground-truth $(x_i)$) to perform signal reconstruction. The method leverages the fact that the statistician has access to $G$ measurement operators, so that she can build ``synthetic'' measurements for each measurement operator using the current estimates $(\hat{x_i})_{i=1}^N$. The introduction of this approach is completed with numerical experiments on image datasets using random (Gaussian) measurement operators, and comparing this approach with a supervised learning approach as well as other simpler algorithms.

The paper contains an interesting and well-written discussion of existing literature: however, not being an expert on the topic of model identification,
I may not be aware of some imporant other works.
The theoretical results of the paper generalize previous works to the case of model identification under binary measurements (both signal reconstruction under binary measurements and model identification under linear measurements were studied before).

**Audience:**

Yes

**Broader Impact Concerns:**

I have no broader impact concern.

**Claims And Evidence:**

Yes

**Requested Changes:**

I will list first the main issues and paths for improvements, before giving a list of more minor points.

**Main issues and improvements --**

1. The acknowledgements mention the funding, which might be a breach of double blindness.
  I will refer to the editor for this point as I am not sure of the importance of this issue.
2. The authors write that the self-supervised algorithm presented in Section 4 is motivated by the analysis of model identification in Section 3. How is that so? Perhaps this would deserve more explanation, especially since the algorithm performs signal reconstruction, not model identification.
3. Why are there so few values of $m$ presented in Figures 6-7-8 ? Adding more values could allows for more convincing plots, e.g. for the scaling with $m$ shown in Figure 6 - right, by taking a large maximal value of $m/n$ (which is at the moment $1.28$).
4. Why did the authors restrict the choice of the trade-off parameter to be $\alpha \in ${0.1, 1, 10}? A more thorough investigation of the optimal choice of $\alpha$ (at least in some particular cases) would be an interesting addition,
  as the current choice seems quite arbitrary.
5. The proofs in appendices contain some mistakes (often easy to fix), and did not always seem to receive the same attention in the writing compared to the main text.While this does not impact the results, the repetition of small mistakes or the lack of clarity can make a bad impression on a careful reader. I list these problems now:
    + In the proof of Theorem 1 the value of $\epsilon$ chosen is (I think) not correct, as one obtains $\delta/\pi - \delta/2 < 0$ in the lower bound between eqs. (34) and (35). This will probably change the constants later on and should be corrected.
    + In the proof of Theorem 6, the equality of eq. (43) should actually be an upper bound, as the vector $v$ does not depend on $g \in [G]$ there is no reason for it to be an equality.
    + I think the derivation of $\delta$ after Theorem 6 should be one of an upper bound on the minimal error $\delta$ achievable for a given value of $m$, the current presentation of a lower bound is a bit confusing. For this minimal error eq. (61) becomes an equality, and the bound of eq. (63) becomes an upper bound, which is consistent (see the next point).
    + I missed something in eq.(63): how does it follow from eq.(62)? Also eq.(62) is only valid if $ae^b \geq 1$, how does this translate to the original parameters?  See also the previous point for a possibly simpler presentation.
    + One should precise for which values of $\xi$ eq.(64) holds (even though it holds even for $\xi$ exponentially small in $n$ I believe).
    + The argument in the proof of Theorem 8 combining Corollary 10 with a union bound needs to be explained more. To me it seems to be using Bernstein's inequality, which would need to be at least mentioned.
    + In the proof of Theorem 8, how does the bound on $Z(S)$ transfers to a bound on $|\Phi(S)|$? This is used in the proof, but not really explained.
    + In the proof of Theorem 8, should the condition $m\sqrt{n} / (sk) > e$ be replaced by (the stronger) $3 m\sqrt{n} / (16sk) > e$?

**Minor points --**

1. In Table 1 one should introduce what the identification error is, since it is only introduced later in the text.
2. In Theorem 1, the condition on $\delta$ should be given before the condition on $m$, since the latter depends on the former.
  Moreover I believe the implication that is proven is $\mathrm{sign}(Ax) = \mathrm{sign}(As) \Rightarrow \|x - s \| < \delta$.
3. Does Section 3.4 assume $k < m$? It seems so since the arguments use binomial coefficients $\binom{m}{k}$, but this condition is not written.
4. Before eq. (25) it would be good to recall that in this setting $T_g$ is a representation of some group under which the set $\mathcal{X}$ is invariant.
5. In Figure 6 (and maybe the latter figures as well), one should emphasize that the PSNR grows logarithmically with $\| x - \hat{x} \|$ as it goes to zero, or write on the axis that the unit is in dB, to enhance readability.
6. In Figure 6 it seems to me that the measurement consistency actually clearly separates from the linear inverse approach for $m/n \gtrsim 0.5$, not exactly at $m/n = 1$ as stated in the text. Is there an explanation for this phenomenon?
7. In Figure 8, one should match the names of the different algorithms in the left and right figures.
8. In the proof of Proposition 3, I think $v$ is just a non-zero element of the nullspace rather than a generator (as the nullspace might have dimension larger than $1$).
9. The bound of eq. (46) is already used in the proof of Theorem 1 without being stated, it should be first stated there.
10. It would be good to add a reference for the standard covering number upper bound above eq. (48).

**Other questions --**
1. Is the bound of Proposition 4 tight (up to constant factors) for some matrices? Proposition 3 shows that if $mG < n$ then it is tight, but what about when $m > n/G$?
2. In Section 3.3, can one obtain a bound using Proposition 2 or Corollary 5? If model identification is difficult then signal recovery should probably also be difficult, but maybe this is not an interesting bound?

**Typos --**
1. Before eq. (27), the sentence ``As the first term...'' needs rewriting.
2. In Figs. 9 and 10, should "each column'' be "each row''?
3. Proof of Theorem 1: "[...] such that'' should be "[...] that''.
4. In eq. (50) the exponent of $\epsilon$ should be $-kG - n$ not $-kG + n$.
5. In eq. (51) a minus sign is missing in front of the log in the denominator.
6. In the proof of Theorem 8, $\mathcal{X}$ should be replaced by $\mathcal{Q}$ in several places.
7. The sentence ``for some $c > 0$'' in the proof of Theorem 8 relates to nothing in the equations.
8. In the proof of Theorem 8, there is a $2$ missing in the exponent in the last equation (when using the previously derived bound on $|\Phi(V_\epsilon(q))|$).

**Strengths And Weaknesses:**

Besides what I have mentioned above in the summary, let me emphasize the following strengths:

1. The paper is well-written and very pleasant to read, and I thank the authors for their effort on this point.
  Results are well described in the introduction, which makes the paper much easier to read.
2. The theoretical bounds presented are quite general and of interest, and well discussed.
3. The self-supervised learning method is a good addition, its precise form is well justified,
  and the numerical analysis showing that it performed comparatively to supervised learning approaches is convincing.

In my opinion, the paper has however some issues which would need corrections and improvements. If these are addressed properly I would recommend the paper for publication. These issues are given in the next paragraph on requested changes.

---

> ### Author Response · Authors · 2023-09-27
> **Response to reviewer DqqG (part 1)**
>
> Thank you for the detailed review, we appreciate the time taken to review our work. We believe that the feedback has helped to improve the paper, in particular the exactness of the proofs. We have uploaded a revised manuscript with changes highlighted in red. We answer your comments in the text below. Since the response does not fit in a single comment, we have divided it into multiple ones.
>
> > The acknowledgments mention the funding, which might be a breach of double blindness. I will refer to the editor for this point as I am not sure of the importance of this issue.
>
> We sincerely apologize for this mistake, we have removed it in the updated manuscript.
>
> > The authors write that the self-supervised algorithm presented in Section 4 is motivated by the analysis of model identification in Section 3. How is that so? Perhaps this would deserve more explanation, especially since the algorithm performs signal reconstruction, not model identification.
>
> Learning a reconstruction function \$f\_{\\theta}\$ from measurement
> data alone has an implicit link with the model identification problem
> since the reconstruction function requires some knowledge about the
> signal set. As discussed in Section 2, the optimal reconstruction
> function
> $$
> \\hat{f}(y) = \\text{centroid}(S\_{y} \\cap \\mathcal{X})
> $$
> where \$S_y\\subset \\mathbb{S}\^{n-1}\$ is the consistency cell
> associated with \$y\$, depends crucially on \$\\mathcal{X}\$. The proposed learning algorithm builds a discrete approximation of the
> signal set using the reconstructed dataset, i.e., \$\\cup\_{g=1}\^G
> \\tilde{\\mathcal{X}}\_g\$ where \$\\tilde{\\mathcal{X}}\_g =
> f\_{\\theta}(\\mathcal{Y}\_g,A\_{g})\$ for \$g=1,\\dots,G\$ and
> \$\\mathcal{Y}\_g\$ is the subset of (training) measurement vectors
> associated with the \$g\$th operator. From a model identification
> perspective, the measurement consistency loss ensures that
> \$\\mathcal{Y}\_g = A_g\\tilde{\\mathcal{X}}\_g\$ for all \$g=1,\\dots,G\$. The second loss ensures consistency across all
> operators, i.e., \$\\tilde{\\mathcal{X}}\_g =
> f\_{\\theta}(A_s\\tilde{\\mathcal{X}}\_g, A_s)\$ for all pairs \$s, g
> \\in \\{1,\\dots,G\\}\$, acting as a proxy for
> \$\\tilde{\\mathcal{X}}\_s = \\tilde{\\mathcal{X}}\_g\$. We have included this discussion in Section 4 of the updated manuscript.
>
>
> > Why are there so few values of $m$ presented in Figures 6-7-8? Adding more values could allow for more convincing plots, e.g. for the scaling with $m$ shown in Figure 6 - right, by taking a large maximal value of $m/n$ (which is at the moment 1.28).
>
> We have included more values of $m$ in Figures 6 and 8 (Figures 6 and 9 in the updated manuscript), showing results for sampling ratios up to $m=4n$. The new plots follow similar trends as the results in the first submission while providing a more detailed understanding of the empirical performance of each method as a function of $m$. In particular, the scaling of the proposed self-supervised method with $m$ shown in Figure 6  is compatible with the $\mathcal{O}(\frac{1}{m}\log m)$ scaling for values of $m>n$.
>
> > Why did the authors restrict the choice of the trade-off parameter to be \{0.1, 1, 10\}? A more thorough investigation of the optimal choice of $\alpha$ (at least in some particular cases) would be an interesting addition, as the current choice seems quite arbitrary.
>
> We empirically observed that all the measurement consistency losses obtained better results for the simple choice $\alpha=1$ except for the logistic loss. We believe this behavior is due to the interplay between the logistic loss and the squared loss used for cross-operator consistency. We included a new experiment in Appendix E of the updated manuscript showing the performance of the proposed method with the logistic loss as a function of the trade-off parameter $\alpha$ and the undersampling ratio $m/n$. We observed that the optimal $\alpha$ decreases with $m$. Thus, in the remaining experiments, we use the simple rule $\alpha=0.1$ for $m<n$ and $\alpha=0.06$ for $m\geq n$.
>
> We believe that a complete theoretical analysis of the optimal choice of $\alpha$ is an interesting but complex problem that is out of the scope of this paper. We leave this analysis for future work.
>
> > The proofs in appendices contain some mistakes (often easy to fix) and did not always seem to receive the same attention in the writing compared to the main text. While this does not impact the results, the repetition of small mistakes or the lack of clarity can make a bad impression on a careful reader. I list these problems now:
>
> We apologize for these mistakes. We have modified the proofs of the main theorems, addressing all of the issues raised by the reviewer. We have also reorganized some parts of the proofs and simplified some parts of the notation with the aim of rendering the proofs more accessible to the reader.

---

> > ### Author Response · Authors · 2023-09-27
> > **Response to reviewer DqqG (part 2)**
> >
> > > In the proof of Theorem 1 the value of $\epsilon$ chosen is (I think) not correct, as one obtains $\delta/\pi - \delta/2$ in the lower bound between eqs. (34) and (35). This will probably change the constants later on and should be corrected.
> >
> > We thank the reviewer for finding this mistake in the choice of $\epsilon$. We have fixed this error in the updated manuscript (see page 20) by choosing $\epsilon = \frac{\delta (4-\pi)}{8+4\pi\sqrt{n\pi/2}}$, which avoids a negative $\delta$ and results in a slight modification of the constants in the final bound:
> >      $$
> >         \textstyle m \geq \frac{4}{\delta} \Big(2k\log\frac{30\sqrt{n}}{\delta} + \log\frac{1}{\xi} \Big)
> >     $$
> >     that holds for $\delta\leq 30 \epsilon_0 \sqrt{n}$.
> >
> > > In the proof of Theorem 6, the equality of eq. (43) should actually be an upper bound, as the vector $v$ does not
> >     depend on $g\in [G]$ there is no reason for it to be an equality.
> >
> > The reviewer is right. Moreover, the inequality was missing the symbol $\exists$ before $v$. We have also simplified and reorganized the proofs of Theorems 1 and 6 (Theorems 1 and 7 in the updated manuscript). We believe they are now easier to understand.
> >
> > > I think the derivation of $\delta$ after Theorem 6 should be one of an upper bound on the minimal error $\delta$ achievable for a given value of $m$, the current presentation of a lower bound is a bit confusing. For this minimal error eq. (61) becomes an equality and the bound of eq. (63) becomes an upper bound, which is consistent (see the next point).
> >
> > We agree with this observation. We have modified the derivation to stress that the upper bound is obtained by choosing the equality in eq. (63) (equation (49) in the updated manuscript). See also the comment below for more details.
> >
> > > I missed something in eq. (63): how does it follow from eq. (62)? Also eq. (62) is only valid if $ae^{b}\geq 1$ , how does this translate to the original parameters? See also the previous point for a possibly simpler presentation.  One should be precise for which values of $\xi$ eq. (64) holds (even though it holds even for exponentially small in $n$ I believe).
> >
> > We have simplified the derivation of $\delta$ in the updated manuscript, avoiding altogether the use of the Lambert function. The new derivation should solve all the issues of the reviewers' comments, while (in our opinion) being easier to follow.
> >
> > > The argument in the proof of Theorem 8 combining Corollary 10 with a union bound needs to be explained more. To me, it seems to be using Bernstein's inequality, which would need to be at least mentioned.
> >
> > We apologize for this as there was indeed a missing explanation in our first derivation. We have now modified the proof of Theorem 8 (Theorem 10 in the updated manuscript) to clarify the use of Bernstein's inequality in the step combining the results from Corollary 10 (Corollary 11 in the updated manuscript) and a union bound over the centers of the covering $Q_{\epsilon}$.
> >
> > Moreover, we have simplified both the statement and the proof of this theorem, introducing explicitly the failure probability. We have also completed this theorem with an additional upper bound on the expectation of the $|\text{sign}(A \mathcal{X})|$ that completes well, we think, the probabilist bound.
> >
> > > In the proof of Theorem 8, how does the bound on $Z(S)$ transfer to a bound on $|\Phi(S)|$? This is used in the proof, but not really explained.
> >
> > Given $\Phi(\cdot) = \text{sign}(A\cdot)$, the number of discontinuities $Z(S)$ can be used to upper bound the number of intersected cells, by noting that $|\Phi(S)|\leq 2^{Z(S)}$ for any set $S\subseteq \mathbb{S}^{n-1}$. This explanation has been included in the updated proof.
> >
> > > In the proof of Theorem 8, should the condition $m\sqrt{n}/(sk) > e$
> >  be replaced by  $3m\sqrt{n}/(16sk) > e$ (the stronger)?
> >
> > Indeed, thank you for spotting this mistake. However, in the simplified proof this bound is not useful anymore. Note that we voluntarily introduced a few crude bounds to simplify further certain constants.
> >
> > > In Table 1 one should introduce what the identification error is, since it is only introduced later in the text.
> >
> > We have included an informal definition of the model identification error $\delta$ in the caption of Table 1 in the updated manuscript, which reads "The identification error $\delta$ corresponds to the maximal error of the optimal estimation of the signal set from binary measurement data alone". The formal mathematical definition is deferred for Section 3 and is encapsulated in Definition 3.1 in the updated manuscript.
> >
> > > In Theorem 1, the condition on $\delta$ should be given before the condition on $m$, since the latter depends on the former.
> >
> > We agree with this observation. We have modified the statements of Theorem 1 and 6 (Theorems 1 and 7 in the updated manuscript) such that the condition on $\delta$ is provided before the one on $m$.

---

> > > ### Author Response · Authors · 2023-09-27
> > > **Response to reviewer DqqG (part 3)**
> > >
> > > > Moreover I believe the implication that is proven is $\text{sign}(Ax)=\text{sign}(Au) \implies \\|x-u\\|\leq \delta$.
> > >
> > > Thank you for spotting this typo. Indeed, the implication of Theorem 1 is $\text{sign}(Ax)=\text{sign}(Au) \implies \\|x-u\\|\leq \delta$.
> > >
> > > > Does Section 3.4 assume $k<m$? It seems so since the arguments use binomial coefficients $\binom{m}{k}$, but this condition is not written.
> > >
> > > Indeed, we assume that $k\leq m$, we have made this assumption explicit in the updated manuscript (see page 11).
> > >
> > > > Before eq. (25), it would be good to recall that in this setting $T_g$ is a representation of some group under which the set $\mathcal{X}$ is invariant.
> > >
> > > Thank you, we have clarified this in the updated manuscript (see page 12).
> > >
> > > > In Figure 6 (and maybe the latter figures as well), one should emphasize that the PSNR grows logarithmically with  $\\|x-\hat{x}\\|$ as it goes to zero, or write on the axis that the unit is in dB, to enhance readability.
> > >
> > > We have modified all the figures displaying a PSNR axis to highlight that the units are in dB.
> > >
> > > > In Figure 6 it seems to me that the measurement consistency actually clearly separates from the linear inverse approach for $m/n\geq 0.5$, not exactly at $m/n=1$ as stated in the text. Is there an explanation for this phenomenon?
> > >
> > > This is an interesting observation. While we do not have a full explanation for this phenomenon, we believe that it might be due to the inductive bias of the convolutional architecture used in the experiments towards clean images. Nonetheless, it is worth noting that the performance of the measurement consistency approach is worse than the proposed SSBM approach, even for values of $m>n$ (see also the point related to the new experiments for more values of $m/n$).
> > >
> > > > In Figure 8, one should match the names of the different algorithms in the left and right figures.
> > >
> > > We have corrected the legends in Figure 8 (Figure 9 in the revised manuscript), such that the names of the algorithms are the same in both subfigures.
> > >
> > > > In the proof of Proposition 3, I think $v$ is just a non-zero element of the nullspace rather than a generator (as the nullspace might have a dimension larger than 1).
> > >
> > > We have modified the proof (see page 8 of the updated manuscript) to mention that $v$ is just an element of the nullspace of $A$ and thus does not necessarily generate the entire subspace.
> > >
> > > > The bound of eq. (46) is already used in the proof of Theorem 1 without being stated, it should be first stated there.
> > >
> > > Thank you for spotting this inconsistency. The bound in terms of Euclidean distance is introduced in the revised manuscript with the statement of Lemma 9 (Lemma 10 of the updated manuscript), and the following proofs of Theorem 1 and 6 (Theorems 1 and 7 of the updated manuscript) directly apply the bound using this distance.
> > >
> > > > It would be good to add a reference for the standard covering number upper bound above eq. (48).
> > >
> > > We have added a reference to *G. Pisier, The volume of convex bodies and Banach space geometry. Cambridge University Press, 1999, vol. 94.* above eq. (48).
> > >
> > > > Is the bound of Proposition 4 tight (up to constant factors) for some matrices? Proposition 3 shows that if $mG<n$ then it is tight, but what about when $mG>n$?
> > >
> > > This is an interesting question. As a special case of Theorem 1 with $mG > n$ measurements and a signal set with box-counting dimension equal to $n$, for an $mG \times n$ Gaussian random matrix $\bar A$, with high probability and by picking the minimal number of required measurements in the condition of this theorem, the largest cell of $\text{sign}(\bar A \cdot)$ has a diameter that decays like $C\frac{n}{mG}$ up to log factors. By a standard boosting argument, it thus means that there exists a $mG \times n$ matrix $\bar A$ with the same consistency cell diameter decay. We have now inserted a remark in page 9 of the updated manuscript to explain this fact.
> > >
> > > Moreover, we would like to stress that (18) in Proposition 3 only provides a necessary condition on the rank of $\bar A$. If we consider a binary matrix $\bar A$ with entries in $\pm 1$, one can show that there is a cell in the related tesselation whose diameter is $\sqrt 2$. This fact is now briefly explained in Example 3.1 of the updated manuscript.

---

> > > > ### Author Response · Authors · 2023-09-27
> > > > **Response to reviewer DqqG (part 4)**
> > > >
> > > > > In Section 3.3, can one obtain a bound using Proposition 2 or Corollary 5? If model identification is difficult then signal recovery should probably also be difficult, but maybe this is not an interesting bound?
> > > >
> > > > The reviewer raises a very interesting point. Consider the optimal reconstruction error of a function learned from measurement data alone, defined as
> > > > $$
> > > >     \hat{f}(y) = \text{centroid}(S_{y} \cap \hat{\mathcal{X}}).
> > > > $$
> > > > where $S_y$ is the consistency cell associated with the measurement vector $y$. This error must be larger than the error of a reconstruction function that has full knowledge about the signal set $\mathcal{X}$. Intuitively, if we have a large model identification error, $\hat{\mathcal{X}}$ will be a bad approximation of $\mathcal{X}$ and thus $\hat{f}$ will obtain large reconstruction errors. The following proposition formalizes this intuition, showing that the reconstruction error of $\hat{f}$ is lower bounded by the model identification error.
> > > >
> > > > **Proposition 8** Given $G$ operators $A_1, \ldots, A_G \in \mathbb{R}^{m \times n}$ and a set $\mathcal{X} \subset \mathbb{S}^{n-1}$ with model identification error equal to $\delta$, there exist points $x_g\in\mathcal{X}$ for $g=1,\dots,G$ such that the reconstruction error is
> > > >     $$
> > > >     \\|\hat{f}( \text{sign}(A_gx_g)) -x_g \\| \geq \delta/2,
> > > >     $$
> > > >     where $\hat{f}$ is the optimal reconstruction function that can be learned from the measurement data $\\{ \text{sign}(A_g\mathcal{X})\\}_{g=1}^{G}$.
> > > >
> > > > *Proof*.  Following the definition of model identification error (Def. 3.1 in the updated manuscript), there exists a point $\hat{x}\in\hat{\mathcal{X}}$ such that $\\|x-\hat{x}\\| \geq \delta$ for all $x\in\mathcal{X}$. According to the construction of the inferred set $\hat{\mathcal{X}}$, there exist some $x_1,\dots,x_{G}\in\mathcal{X}$ such that $\text{sign}(A_g\hat{x})=\text{sign}(A_gx_g)$ for all $g=1,\dots,G$. Therefore, for any $g\in\{1,\dots,G\}$, the diameter of the set $S_{\text{sign}(A_gx_g)} \cap \hat{\mathcal{X}}$ is at least  $\\|x_g-\hat{x}\\|$ since we have that both $x_g$ and $\hat{x}$ belong to this set. As the optimal reconstruction function outputs the centroid of the cell, the reconstruction error of the point $x_g$ is at least $\\|x_g-\hat{x}\\|/2 \geq \delta/2$. $\\square$
> > > >
> > > >
> > > > Therefore, we can use the results on model identification developed in Section 3.1 to lower bound the reconstruction error for the case where the function is learned from measurement data only. In particular, combining this result with Corollary 5 (Corollary 6 in the revised manuscript), we obtain that the (worst-case) reconstruction error should be larger than $\frac{1}{3}\frac{n}{mG}$. It is worth noting that this result also holds for the case where we have a single operator and group invariance, i.e., when $A_g=AT_g$ for $g=1,\dots,G$.
> > > >
> > > > We have included this proposition and discussion in the updated manuscript (see pages 10 and 11).
> > > >
> > > > > Before eq. (27), the sentence ``As the first term...'' needs rewriting.
> > > >
> > > > We have fixed this sentence in the updated manuscript.
> > > >
> > > > > In Figs. 9 and 10, should "each column'' be "each row''?
> > > >
> > > > No, the image of each column is observed via a different operator $A_g$ (one out of the $G$ possible ones). We have modified the caption of these figures in the updated manuscript to make this point more clear.
> > > >
> > > > > Proof of Theorem 1: "[...] such that'' should be "[...] that''.
> > > >
> > > > Thank you. We have fixed this in the updated manuscript.
> > > >
> > > > > In eq. (50), the exponent of $\epsilon$ should be $-kG-n$ not $-kG+n$.
> > > >
> > > > Thank you for spotting this typo, we have corrected it in the updated manuscript.
> > > >
> > > > > In eq. (51) a minus sign is missing in front of the log in the denominator.
> > > >
> > > > This was indeed an error in our proof; thank you for finding it. We have now clarified this development.
> > > >
> > > > > In the proof of Theorem 8, $\mathcal{X}$ should be replaced by $Q$ in several places.
> > > >
> > > > Indeed. This is now corrected.
> > > >
> > > > > The sentence ``for some $c>0$'' in the proof of Theorem 8 relates to nothing in the equations.
> > > >
> > > > We have removed this mistake in the updated manuscript.
> > > >
> > > > > In the proof of Theorem 8, there is a 2 missing in the exponent in the last equation (when using the previously derived bound on  $|\Phi(V_\epsilon(q))|$).
> > > >
> > > > Thank you for spotting this mistake. As explained above, the proof of this theorem has been updated and this previous (erroneous) bound is not useful anymore.

---

### Decision · Action_Editors · 2023-10-11

**Recommendation:** Accept as is

**Comment:**

This paper submitted to TMLR has received unanimous approval from reviewers for its novel and pertinent contributions. The authors have been in particular commended for their response to feedback, providing comprehensive explanations and making corrections that have enhanced the quality of the paper.

**Audience:**

The paper is deemed highly appropriate for TMLR publication by the referees.

**Claims And Evidence:**

The paper delves into the issue of one-bit compressed sensing, observing a one-bit quantization of a signal post its passage through a sensing matrix. Unlike prior studies that aimed at reconstructing the signal given the knowledge of the signal set, this research emphasizes reconstructing the signal set itself using multiple observations. The authors present both upper and lower bounds for this reconstruction, which align closely if the set has a low box-counting dimension. The paper also broadens previous findings on signal reconstruction from sparse settings to cases with low box-counting dimensions. Additionally, a self-supervised learning loss for the signal is introduced, supported by experiments comparing it to other methods.

The paper is of high quality, showcasing novel and rigorously proven results that cater to both theoretical scholars and practitioners in the realm of self-supervised learning in compressed sensing. The content is accessible, even for those unfamiliar with the subject, and is complemented by illustrative diagrams that simplify binary recovery concepts.